# Maternal transmission of bacterial microbiota during embryonic development in a viviparous lizard

Nina Montoya-Ciriaco,[1] Stephanie Hereira-Pacheco,[2] Arturo Estrada-Torres,[2] Luc Dendooven,[3] Fausto R. Méndez de la Cruz,[4] Elizabeth Selene Gómez-Acata,[5] Aníbal H. Díaz de la Vega-Pérez,[6] Yendi E. Navarro-Noya[5]

**ABSTRACT** The maternal transmission of microbiota during embryonic development of vertebrates is still poorly understood. Here, we used high-throughput sequencing of the 16S rRNA bacterial genes to determine the bacterial communities in the gastrointestinal tract and amniotic environment, i.e., the amniotic fluid, amniotic membrane and extraembryonic yolk, of embryos at the last stage of development of the viviparous lizard *Sceloporus grammicus* Wiegmann, 1828. We compared these communities to those found in the maternal intestine, mouth, cloaca, and the aseptic ventral skin as a control of the aseptic technique. Our results showed that bacterial 16S rRNA genes were present in the embryos of *S. grammicus*. Their diversity was lower and more similar in composition between individuals than those found in the maternal tissues. This suggests that a strong control exists on the transmission of bacteria from the mother to the embryos. We found 78% of the embryonic amplicon sequence variants (ASVs) in the maternal bacterial microbiota, suggesting that the transmission of bacteria from the mother to the embryos is a continuous process and some bacteria may have been transferred during early embryonic stages. The embryonic bacteria were found to overlap mostly with those found in the mouth and aseptic ventral skin of the mother, although it is difficult to conclude that the shared ASVs originated from these maternal tissues. Our study provides evidence of microbiota vertical transfer during embryonic development in the animal kingdom. It also highlights that this maternal transmission could be included in the maternal effects that impact the offspring.

**IMPORTANCE** We investigated the presence and diversity of bacteria in the embryos of the viviparous lizard *Sceloporus grammicus* and their amniotic environment. We compared this diversity to that found in the maternal intestine, mouth, and cloaca. We detected bacterial DNA in the embryos, albeit with a lower bacterial species diversity than found in maternal tissues. Most of the bacterial species detected in the embryos were also found in the mother, although not all of them. Interestingly, we detected a high similarity in the composition of bacterial species among embryos from different mothers. These findings suggest that there may be a mechanism controlling the transmission of bacteria from the mother to the embryo. Our results highlight the possibility that the interaction between maternal bacteria and the embryo may affect the development of the lizards.

**KEYWORDS** early life microbiome, Is microbiome inherited from the mother?, maternal effects, maternal microbiome, microbial transmission, reptile microbiome

Address correspondence to Yendi E. Navarro-Noya, yendiebenezer.navarro.n@uatx.mx, or Aníbal H. Díaz de la Vega-Pérez, anibal.helios@gmail.com.

The authors declare no conflict of interest.

See the funding table on p. 18.

Animal development might rely on microbial signals that can result from transient interactions or persistent colonization by microorganisms, or from microbial molecular cues (1). Throughout animal ontogeny, microbial communities and/or their molecular signals are integrated into development programs (2–4). For instance, studies

on gnotobiotic mice have shown that the microbiota provides developmental cues, as myeloid cell function and gastrointestinal (GI) differentiation are reduced in these animals (3, 5). In fish, the gastrointestinal microbiota has been found to initiate cell division in intestinal stem cells through the β-catenin signaling pathway (6). In rodents, bacterial metabolites released early in life have been shown to influence brain development and the blood-brain barrier (7, 8). These microbial symbionts can be acquired vertically (from the host's parents) or horizontally (from the environment, other host species, or other members of the same species) (1, 9).

The vertical transmission of bacterial microbiota can occur through various mechanisms in the animal kingdom. In some cases, such as with sap-feeding insects (10, 11) and cockroaches (12), bacteria are transmitted via intracellular infection of oocytes, i.e., transovarial transmission. Other mechanisms include early inoculation during seed formation, egg laying, or passage through the birth canal (13–15). The mechanisms for vertical transmission have been extensively investigated in invertebrates, such as insects, nematodes, sponges, clams, and oysters (16–19). However, research on vertical transmission in vertebrates, especially other than Mammalia, has been limited. In non-mammalian animals, research has mainly focused on maternally transmitted pathogens in oviparous species, mainly in poultry and fish. For instance, the transovarial transmission of pathogens in the egg yolk has been reported to occur in laying hens (20), chickens (21), and salmonid fish (22). In wild animals, a study with river turtles, *Podocnemis expansa* and *P. unifilis*, identified several Enterobacteriaceae species in the eggs but not in the turtle nests (23), suggesting a possible maternal origin. In another study, *Pseudomonas*, *Salmonella*, *Enterobacter*, and *Citrobacter* were isolated from the shell, albumen, and yolk of eggs just hatched from the cloaca of green turtles (*Chelonia mydas*) (24). In the little skate, *Leucoraja erinacea*, the bacterial microbiota on the skin, gills, and egg capsule were studied six times during ontogeny, revealing a highly diverse bacterial community on the inner surface of the egg capsule that had the same composition as the adult bacteria but a different structure (25). Also, bacterial communities in the eggs of *Sceloporus undulatus* lizards differ from those in their environment (15). To date, there has been no comprehensive study on the maternal transmission of bacterial microbiota during embryonic development in non-mammalian viviparous vertebrates, and the anatomical maternal origin of these microorganisms has yet to be determined.

There are several proposed routes for maternal transfer of bacteria in mammals, including: (i) the migration of maternal vaginal bacteria to the fetal environment from the female reproductive tract, (ii) the hematogenous spread of maternal oral bacteria to the placenta, and (iii) the translocation of maternal intestinal bacteria to the fetus through dendritic cells and intestinal epithelium absorption (26, 27). However, these routes remain largely unexplored in non-mammalian viviparous vertebrates.

The aim of our study was to investigate bacterial communities in the gastrointestinal tract of *S. grammicus* embryos, amniotic tissues (i.e., amniotic fluid, amniotic membrane, and extraembryonic yolk), and maternal intestine, mouth, and cloaca. We aimed to address three questions. (i) Is there maternal transmission of bacteria during embryonic development in viviparous lizards? (ii) What are the taxonomic identity and potential functions of these bacteria? and (iii) What is the possible maternal origin of these bacteria?

Viviparous lizards are an excellent model for studying the processes that lead to placentation in vertebrates (28). The morphological and functional changes in the placentas of lizards with different nutritional strategies provide compelling evidence for the evolution of viviparity, which may involve mechanisms of microbiota transmission. Our study aimed to broaden knowledge about the maternal microbiota transmission and to better our understanding of the functions of the gastrointestinal microbiota in its host, such as the possible role played during embryonic development.

## RESULTS

### Taxonomic diversity of embryonic and maternal bacterial microbiota

A total of 65 samples from eight pregnant females of *S. grammicus* in the last stage of development (29) were analyzed through metabarcoding of the bacterial communities using the V3-V4 regions of the 16S rRNA gene and MiSeq 2 × 300 runs, i.e., maternal small intestine ($n = 5$), and swabs from the mouth ($n = 8$), cloaca ($n = 8$), and the ventral body region after the aseptic technique ($n = 8$) (see Materials and Methods; hereafter aseptic ventral skin). In addition, the amniotic fluid ($n = 9$), embryonic GI tract ($n = 17$), amniotic membrane (hereafter membrane; $n = 6$), and extraembryonic yolk (hereafter yolk; $n = 4$) of one or two embryos per mother lizard were analyzed (Fig. 1). Sequencing produced a total of 766,270 reads (11,794 ± 10,529 in average per sample) and 780 amplicon sequence variants (ASVs) (Table S1).

We used Hill numbers to analyze alpha diversity, which accounts for the effective number of ASVs, and weight the abundance of each ASV with the order of diversity $q$ ($q = 0$ represents richness or observed ASVs, $q = 1$ frequent or typical ASVs, and $q = 2$ dominant ASVs). In general, the maternal samples had a significantly higher diversity than embryonic samples [considering richness ($q = 0$) and frequent ASVs ($q = 1$); $P < 0.05$; Table S2). The bacterial richness ($q = 0$) in the maternal samples was similar in the studied sections ($P > 0.05$), while in the embryonic samples, it was different in the following

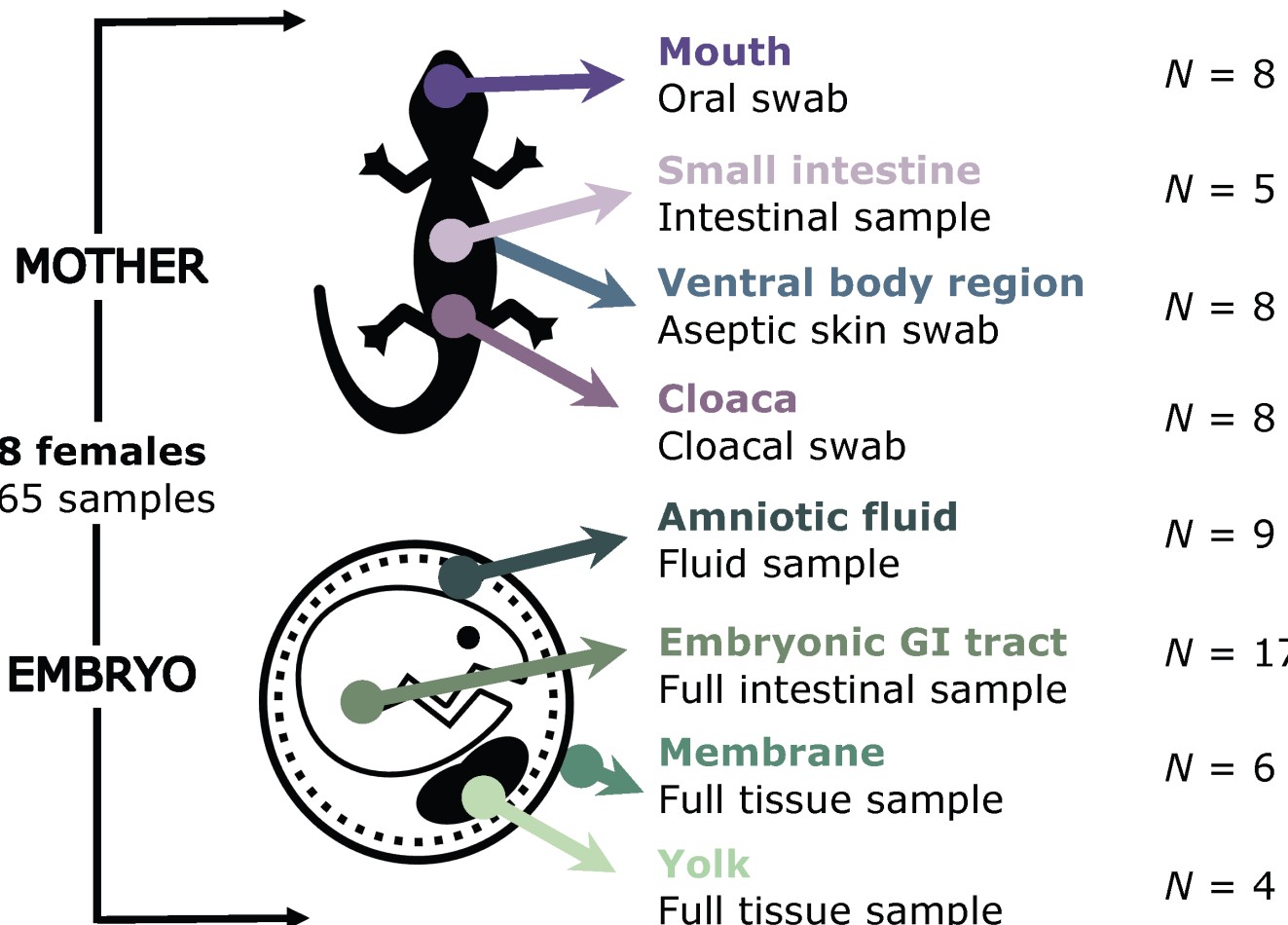

**FIG 1** Samples collected from eight pregnant females of *Sceloporus grammicus* Wiegmann, 1828. They were taken from the maternal mouth, small intestine, cloaca, and aseptic control from the ventral body region (aseptic ventral skin), as well as the amniotic fluid, gastrointestinal tract, membrane, and yolk from each embryo. All samples were processed individually under strict sterile conditions to prevent contamination. Only samples that were not cross-contaminated were included in the study. Two embryos from each female were selected for sequencing of the 16S rRNA gene.

order: embryonic GI tract (33 ± 7) >amniotic fluid (31 ± 9) and membrane (21 ± 2) >yolk (20 ± 3) ($P < 0.05$; Table 1). The bacterial diversity in the maternal sections considering the frequent ($q = 1$) and dominant ($q = 2$) ASVs was higher in the small intestine than in the mouth and cloaca. In the embryo, the diversity of frequent ASVs was higher in the GI tract than in the yolk, while the diversity of dominant ASVs was similar in the embryonic sections (Table 1). Although we used an aseptic technique to disinfect the dissection region on the ventral body region, some bacteria remained resulting in 45 ± 9 total ASVs.

## Taxonomic composition of embryonic and maternal bacterial microbiota

The taxonomic composition of the embryonic and amniotic samples (amniotic fluid, embryonic GI tract, membrane, and yolk) was similar and dominated by the phyla Proteobacteria and Firmicutes (Fig. 2A). At a lower taxonomic level, the classes Alphaproteobacteria and Gammaproteobacteria, and the families Xanthomonadaceae and Comamonadaceae were the most abundant in the embryo and amniotic environment (Fig. 2B and C). In contrast, the taxonomic composition of the maternal samples varied between the different sites. The maternal small intestine was dominated by Firmicutes, followed by Proteobacteria, Actinobacteriota, and Bacteroidota, the mouth by Proteobacteria, Campylobacterota, and Bacteroidota, and the cloaca by Campylobacterota and Proteobacteria (Fig. 2A). The relative abundance of Verrucomicrobiota and Desulfobacterota was <1% in the maternal samples. The small intestine was dominated by the classes Gammaproteobacteria, Clostridia, Bacteroidia, and Actinobacteria, and the families Lachnospiraceae and Pseudomonadaceae. The bacterial community in the mouth was dominated by the classes Gammaproteobacteria and Campylobacteria, and the families Comamonadaceae, Helicobacteraceae, and Xanthomonadaceae, while in the cloaca by Campylobacteria, Gammaproteobacteria, Helicobacteraceae, and Hafniaceae (Fig. 2B and C). The aseptic ventral skin was dominated by the phylum Proteobacteria (Fig. 2A), the class Gammaproteobacteria (Fig. 2B), and the families Comamonadaceae, Pseudomonadaceae, and Xanthomonadaceae (Fig. 2C).

**TABLE 1** Hill numbers of the bacterial communities in the mouth, cloaca, small intestine, and aseptic ventral skin (aseptic control) of female pregnant *Sceloporus grammicus* Wiegmann, 1828, and the amniotic fluid, embryonic gastrointestinal tract, membrane, and yolk of 40 stage embryos

| Sample | Location | Effective number of amplicon sequence variants | | |
|---|---|---|---|---|
| | | $q = 0$ | $q = 1$ | $q = 2$ |
| Mother | Mouth | 58.0 ± 60.5 | 8.5 ± 12.1a | 3.9 ± 4.5a |
| | Cloaca | 41.0 ± 14.8 | 6.4 ± 4.4a | 4.5 ± 3.6a |
| | Small intestine | 71.4 ± 61.3 | 26.0 ± 17.5b | 15.6 ± 11.2b |
| | Aseptic ventral skin | 44.5 ± 8.7 | 5.6 ± 0.7a | 3.1 ± 0.5a |
| | *F* value[a] | 0.593 | 4.522 | 5.349 |
| | *P* value | 0.627 | **0.015** | **0.008** |
| | *P* value (perm)[b] | 0.703 | **0.016** | **0.006** |
| Embryo | Amniotic fluid | 31.1 ± 8.7ab | 4.9 ± 0.5ab | 3.4 ± 0.3 |
| | Embryonic GI tract | 32.8 ± 7.4b | 5.6 ± 1.1a | 3.6 ± 0.5 |
| | Membrane[c] | 20.7 ± 1.7a | 4.6 ± 0.4ab | 3.4 ± 0.3 |
| | Yolk | 20.0 ± 3.0a | 4.3 ± 0.7b | 3.1 ± 0.4 |
| | *F* value[d] | 6.535 | 3.602 | 1.562 |
| | *P* value | **0.006** | **0.043** | 0.245 |
| | *P* value (perm)[e] | **0.004** | **0.036** | 0.285 |

[a]Linear mixed effects model with maternal identity as a random factor was used to test the significant differences.
[b]Linear mixed effects model with 1,000 Monte-Carlo permutations and maternal identity as a random factor was used to test the significant differences.
[c]External membrane that contains the embryo.
[d]Linear mixed effects model with embryo identity as a random factor was used to test the significant differences.
[e]Linear mixed effects model with 1,000 Monte-Carlo permutations and embryo identity as a random factor was used to test the significant differences.

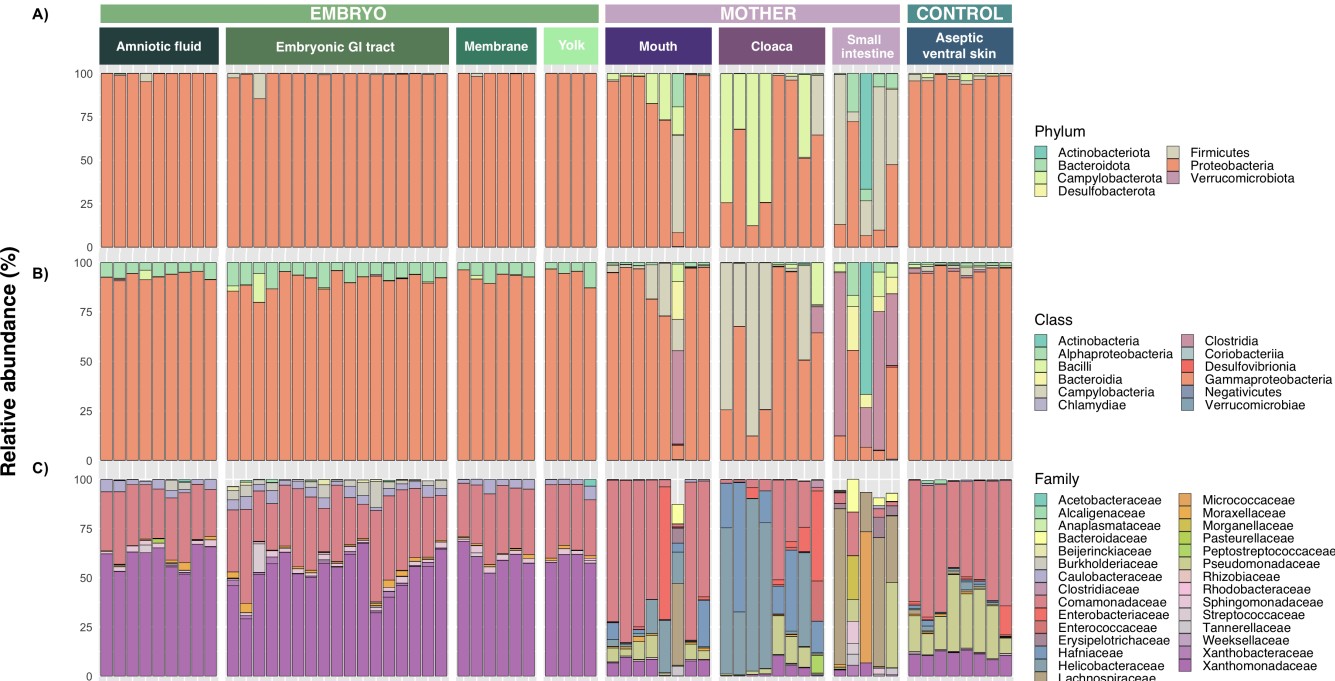

**FIG 2** Barplots with the relative abundance (%) of the bacterial communities at different taxonomic levels in the amniotic fluid, embryonic gastrointestinal tract, the amniotic membrane (membrane), and the yolk of the 40 stage embryos (embryonic samples), and the mouth, cloaca, small intestine, and aseptic ventral skin (aseptic control) of pregnant females of *Sceloporus grammicus* Wiegmann, 1828 (maternal samples). (A) Taxonomic level of phylum, (B) class and (C) family.

## Community composition and structure of ASVs of embryonic and maternal bacterial microbiota

The most abundant ASV in all maternal and embryonic samples belonged to *Curvibacter lanceolatus* (ASV 1) (Fig. 3). Amplicon variants 2, 8, 13, 15, and 43, abundant in all samples except for the maternal small intestine, were identified as belonging to *Stenotrophomonas*, *Brevundimonas,* and *Curvibacter*. A cluster of ASVs exclusive to the embryonic GI tract was identified as members of *Ralstonia* (ASV 10), *Bradyrhizobium* (ASV 21), and *Bosea* (ASV 45). The maternal samples also showed unique clusters, such as *Pseudomonas* (ASV 5), *Variovorax* (ASV 12), and *Stenotrophomonas rhizophila* (ASV 18), that were highly abundant in the maternal mouth. In the maternal cloaca, the most abundant ASVs were *Helicobacter* (ASV 4), *Hafnia paralvei* (ASV 7), and *Hafnia* (ASVs 14 and 16). Notably, ASV 4 (*Helicobacter*) was more abundant in the maternal cloaca compared to all other samples. Several ASVs, including members of Enterobacteriaceae (ASVs 9, 20, 31, 35, 37, and 42) were shared between the maternal mouth and cloaca. The maternal small intestine harbored a group of ASV members of Lachnospiraceae (e.g., ASVs 23, 29, 40, 34, and 48), with ASVs 33 and 41 identified as *Lachnoclostridium* and *Hungatella*. The aseptic ventral skin shared several ASVs with other epithelia, such as the oral and cloacal mucosa, including ASVs 4, 5, 12, and 18.

A principal component analysis (PCA) revealed two major clusters. The embryonic samples formed a cluster on the right very close to each other indicating small variations in the bacterial structure of the embryonic samples, while the maternal samples formed a more dispersed cluster on the left. The cluster of embryonic samples is separated from the aseptic ventral skin, indicating that the bacterial communities of embryos are different from the bacterial communities of the aseptic control (Fig. 4). The permutational multivariate analysis of variance (perMANOVA) using Aitchison distances showed a significant difference between the structure of embryonic and maternal bacterial communities, specifically between the amniotic environment, embryonic GI tract, and aseptic ventral skin (aseptic control), which explained the 66% of the total variance (*F*

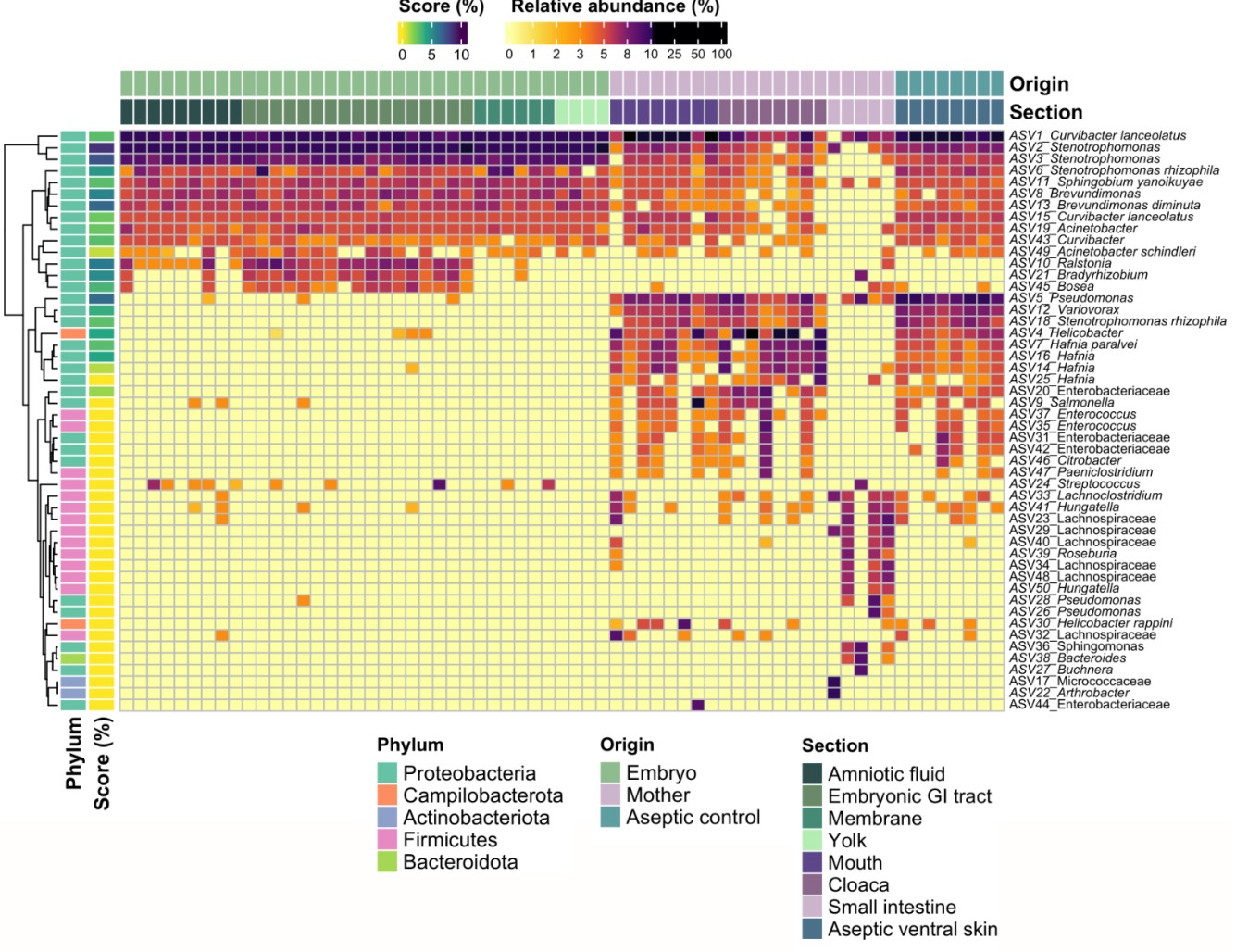

**FIG 3** Heatmap with the 50 most abundant bacterial amplicon sequence variants (%) in the mouth, cloaca, small intestine (maternal samples), aseptic ventral skin (aseptic control) of pregnant female *Sceloporus grammicus* Wiegmann, 1828, and the amniotic fluid, embryonic gastrointestinal tract, membrane, and yolk of 40 stage embryos (embryonic samples). Score indicates the importance of each bacterial ASV in a supervised learning model calculated with q2-sample-classifier.

= 16.11, $P < 0.001$, $R^2 = 0.66$). The ASV 3, belonging to the genus *Stenotrophomonas*, was enriched in embryonic samples, specifically in the embryonic GI tract. The ASV 23, belonging to the Lachnospiraceae family, was enriched in the maternal small intestine, while the ASVs 4 (*Helicobacter*) and 16 (*Hafnia*) were enriched in the maternal cloaca (Fig. 4). The Aitchison distances among embryonic samples were significantly lower than those among maternal samples ($P < 0.001$; Fig. S1).

## Potential reagent contamination

Upon reviewing the literature on the bacterial communities in environments with low biomass, we came across a study that analyzed potential reagent contaminants in the meconium of premature babies (30). The study identified *Curvibacter lanceolatus* as a potential reagent contamination (31). Given that the most abundant and prevalent ASV in our study was identified as *C. lanceolatus*, we decided to apply the same analysis to check for possible reagent contaminants in our samples, although none of our reagent controls were tested positive (data not shown).

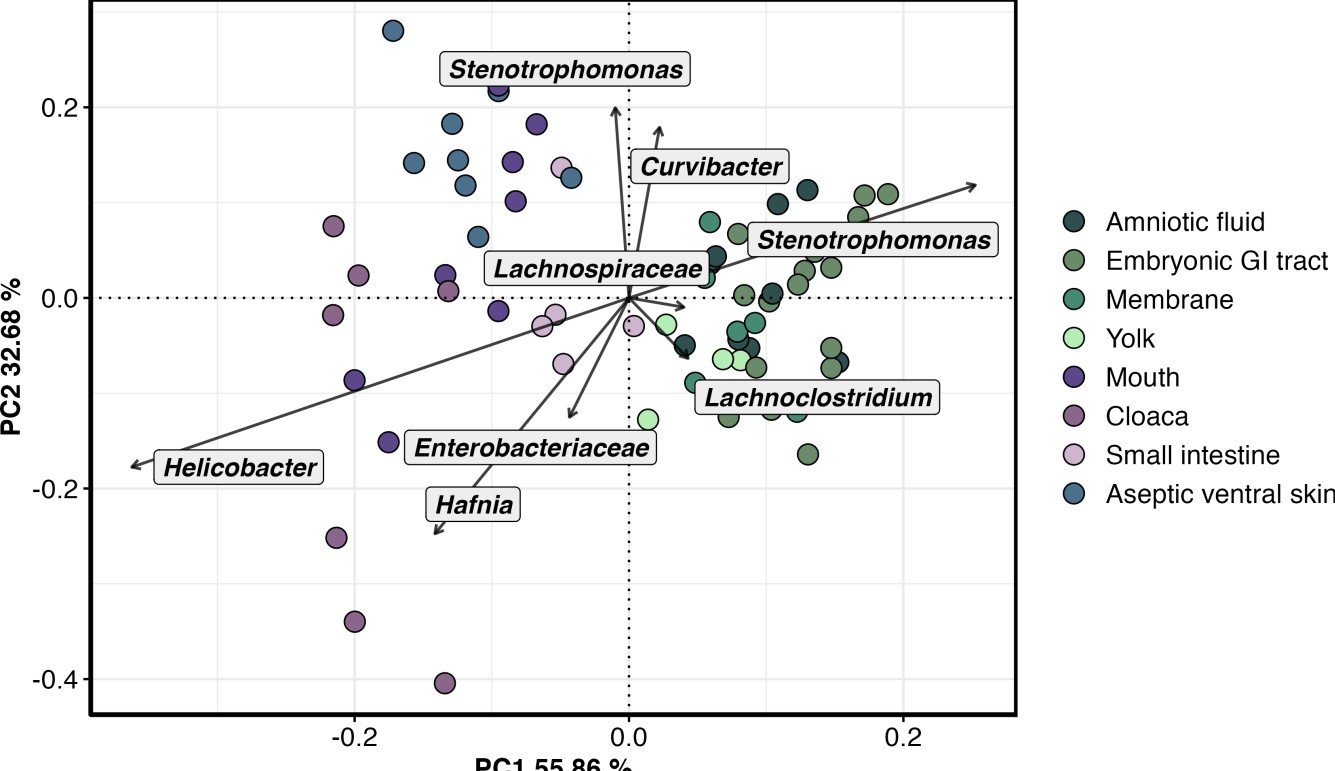

**FIG 4** Principal component analysis of the bacterial community structure at the amplicon sequence variant level in the mouth, cloaca, small intestine (maternal samples), aseptic ventral skin (aseptic control) of female pregnant *Sceloporus grammicus* Wiegmann, 1828, and the amniotic fluid, embryonic gastrointestinal tract, membrane, and yolk of 40 stage embryos (embryonic samples).

We identified five ASVs as possible contaminants: *Eubacterium* group, *Parabacteroides*, *Lachnoclostridium*, Ruminococcaceae, and Lachnospiraceae. However, none of these ASVs were detected exclusively in the embryonic samples.

## Functional prediction of embryonic and maternal bacterial microbiota

The Phylogenetic Investigation of Communities by Reconstruction of Unobserved States (PICRUSt) analysis was used to predict the potential functions of the bacterial microbiota in the maternal and embryonic samples. Although this type of analysis is merely a prediction based on ancestral phylogenetic reconstruction, it is generally found helpful for a broad panorama of the functional potential in a bacterial community (32). The most abundant functions in the embryonic samples were related to aromatic compounds and amino acids degradation, e.g., leucine and 4-aminobutanoate degradation, pyruvate fermentation, and NAD and heme biosynthesis. The most abundant functions in maternal tissues were related to cofactor, carrier and vitamin biosynthesis, fermentation, and carbohydrate degradation, such as fucose and rhamnose degradation, menaquinol and demethylmenaquinol biosynthesis, dTDP-N-acetylthomosamine biosynthesis, besides pyruvate and succinate fermentation (Fig. 5). The CMP-legionaminate (5,7-diacetamido-3,5,7,9-tetradeoxy-D-*glycero*-D-*galacto*-nonulosonic acid) biosynthesis I was exclusive in the embryonic and maternal GI tract, while other functions exclusively found in the embryonic GI tract were involved in the degradation of aromatic compounds, such as mandelate, arabinose, and catechol.

## Overlap of embryonic and maternal bacterial microbiotas

We compared the ASVs present in embryonic samples with those found in maternal samples. We found that 78% (83 out of 106) of ASVs detected in embryonic tissues were

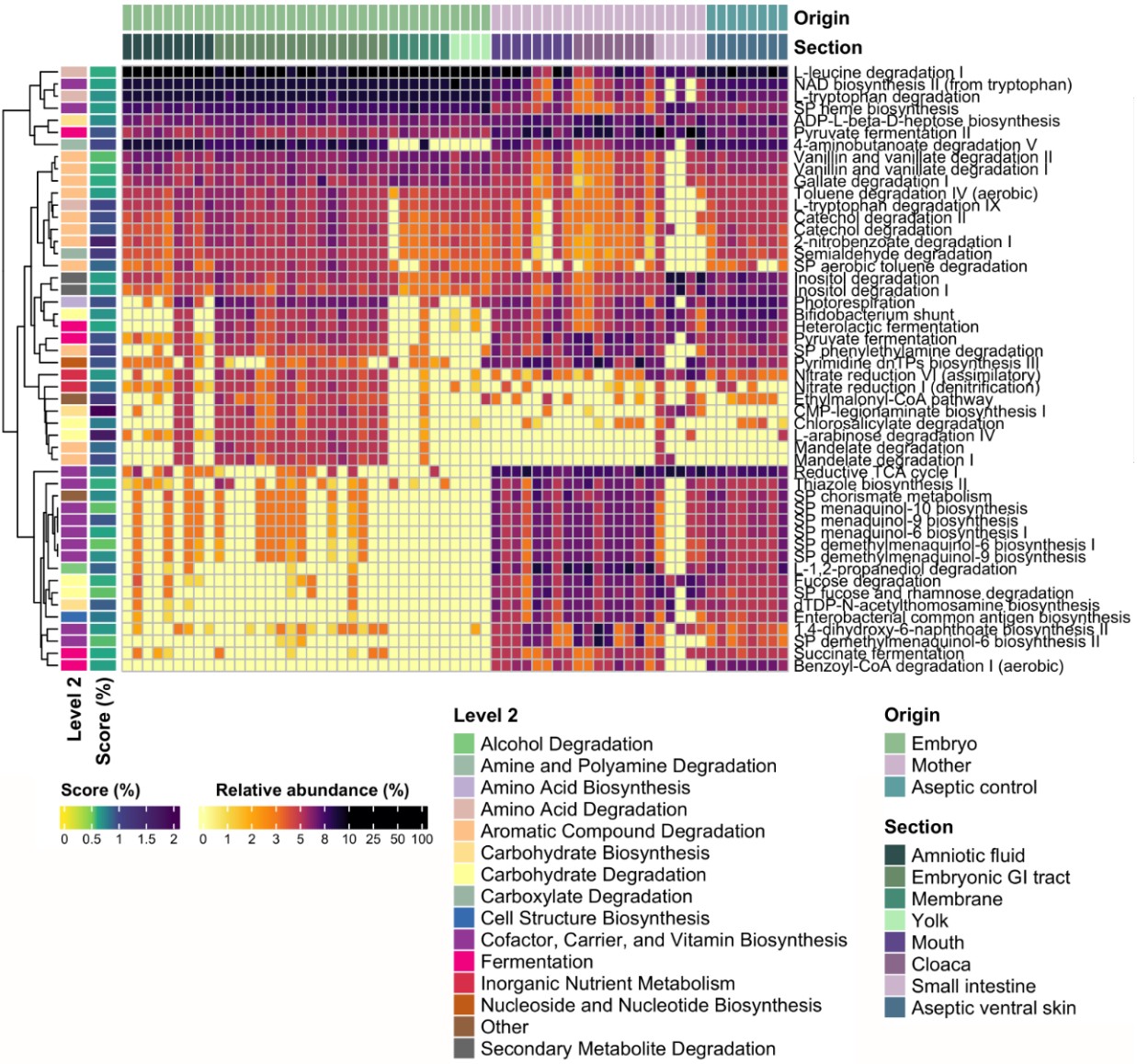

**FIG 5** Heatmap of the putative bacterial functions (%) in the mouth, cloaca, small intestine (maternal samples, purple), aseptic ventral skin (aseptic control, blue) of female pregnant *Sceloporus grammicus* Wiegmann, 1828, and the amniotic fluid, embryonic gastrointestinal tract, membrane, and yolk of the 40 stage embryos (embryonic samples, green). Score indicates the importance of the putative bacterial functions in a supervised learning model calculated with q2-sample-classifier. The top 50 putative functions with the highest scores were selected for plotting. SP, superpathway.

also present in maternal tissues (Fig. 6A). The Jaccard dissimilarity was 0.67 which means that the non-shared species between maternal and embryonic samples represented 67% of total species. The partition analysis of Jaccard dissimilarity into nestedness and turnover (33) was used to assess the degree to which the embryonic communities were a subset of the maternal microbiota. Distance metrics revealed that 0.49 (73% of the total dissimilarity) was due to nestedness, while 0.18 (27% of the total dissimilarity) was due to species replacement.

Comparing shared ASVs in the embryonic and maternal samples showed a significant overlap. The amniotic fluid shared 55% (45 of 82) of its ASVs with the maternal mouth, 44% (36 of 82) with the cloaca, 42% (35 of 82) with the small intestine, and 47% (39 of 82) with aseptic ventral skin (Fig. 6B). Additionally, 20% of ASVs from amniotic fluid were shared with all maternal samples (mouth, cloaca, and small intestine) and 33% with all epithelial samples (mouth, cloaca, and aseptic ventral skin). The embryonic GI tract

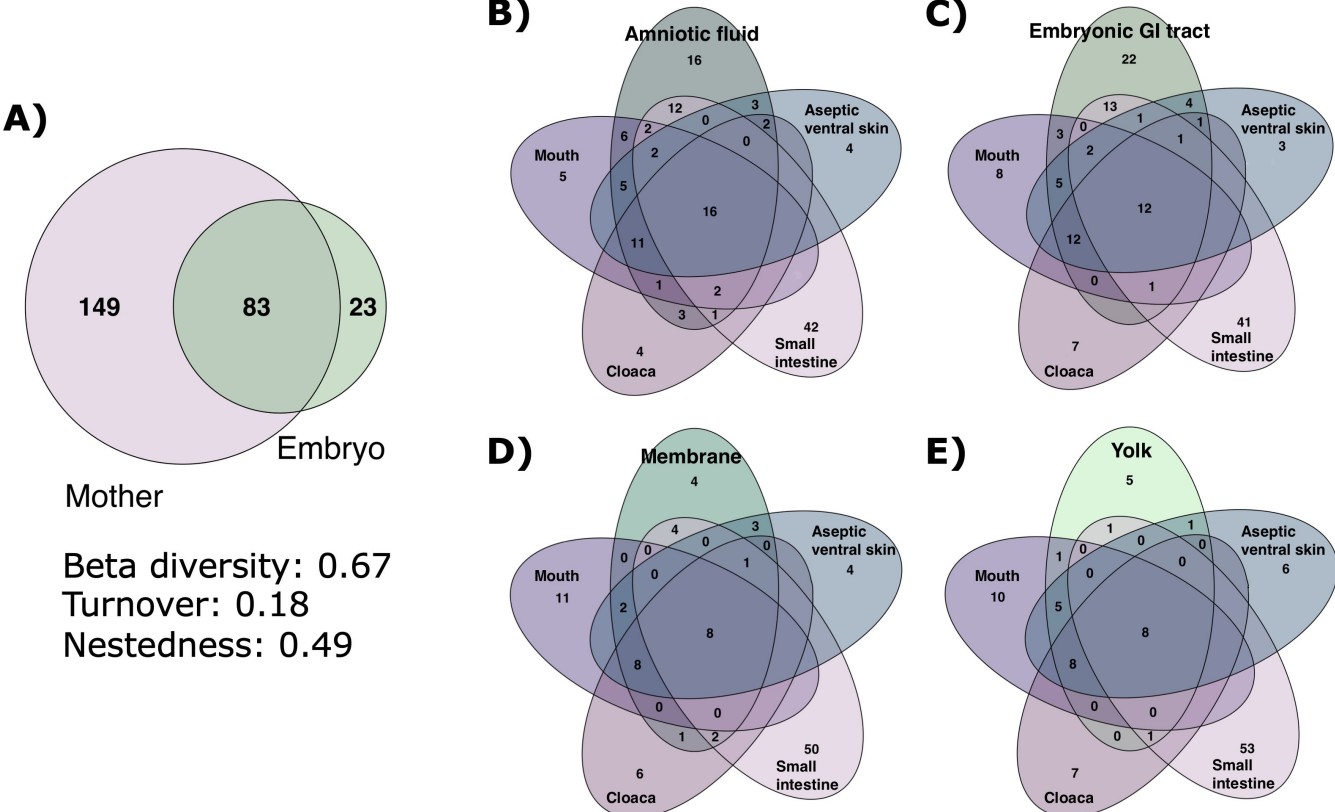

**FIG 6** Bacterial amplicon sequence variants shared in the mouth, cloaca, small intestine, aseptic ventral skin (aseptic control) of female pregnant *Sceloporus grammicus* Wiegmann, 1828, and the amniotic fluid, embryonic gastrointestinal tract, membrane, and yolk of 40 stage embryos. (A) Venn diagram of exclusive and shared ASVs in maternal tissues and embryos. (B–E) Venn diagrams of exclusive and shared ASVs in maternal tissues and amniotic fluid, embryonic GI tract, membrane, and yolk, respectively.

shared 40% (31 of 78) of its ASVs with the maternal intestine (16% of ASVs were shared in the two intestines), 43% with the mouth, 36% with the cloaca, and 49% with aseptic ventral skin (Fig. 6C). Moreover, 15% of the ASVs from the embryonic GI tract were shared among all maternal samples, 30% with maternal epithelia, and 28% were not shared with any maternal sample. The yolk and membrane had the smallest communities, and almost all their ASVs overlapped with those in maternal tissues (88% of yolk and 83% of membrane; Fig. 6D and E). Specifically, the yolk shared 73% (22 of 30) of its ASVs with the maternal mouth, 56% with the cloaca, 30% with the intestine, and 73% with ventral skin, while the membrane shared 54% (18 of 33) with the maternal mouth, 60% with the cloaca, 45% with intestine, and 66% with the aseptic ventral skin.

Comparing only the ASVs found in the embryonic samples with the tissues of their corresponding mother resulted in an overlap that ranged from 4% to 76% (Fig. 7). The highest overlapping was detected between maternal mouth and ventral skin with embryonic tissues, while the lowest was detected between maternal intestine and the embryonic tissues. The intraindividual variation in the embryonic samples was low while high in the maternal samples (Fig. S1). The perMANOVA showed a significant difference between the embryonic and maternal bacterial communities ($F = 7.78$, $P < 0.001$, $R^2 = 0.61$).

We determined the possible maternal origin of bacteria in the embryos with FEAST (fast expectation-maximization for microbial source tracking) analysis. Overall, we found high source contributions of maternal mouth (mean 28.2%) and aseptic ventral skin (mean 27.8%), and from unknown sources (mean 45.4%; Fig. S2). Maternal cloaca and small intestine were not significant potential sources of bacteria. Over half of

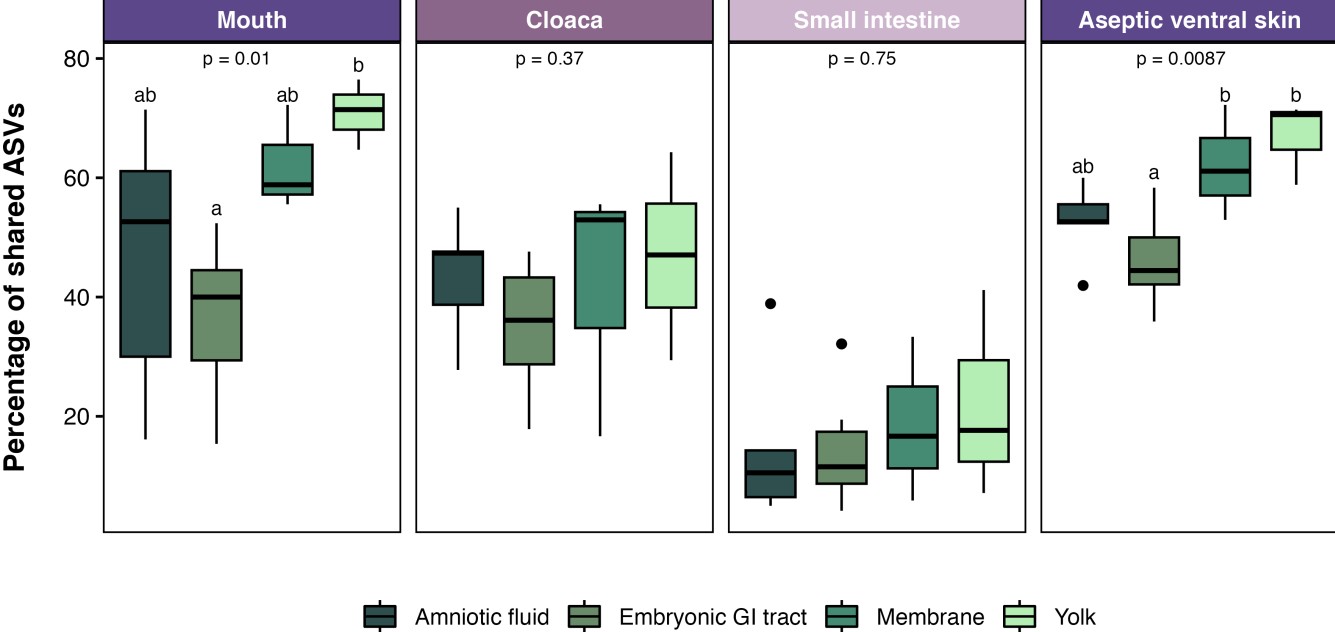

**FIG 7** Shared amplicon sequence variants in each lizard mother and its embryos. Maternal samples included mouth, cloaca, small intestine, and aseptic ventral skin (as an aseptic control) of female pregnant *Sceloporus grammicus* Wiegmann, 1828, and the amniotic fluid, embryonic gastrointestinal tract, membrane, and yolk of 40 stage embryos.

the bacterial composition in the embryonic GI tract (54.8%) originated from unknown sources. This result is consistent with the significant presence of unique ASVs in the embryonic GI tract.

## DISCUSSION

The maternal transmission of microbiota during embryonic development is still little explored in vertebrate embryos. In this study, we determined the diversity of 16S rRNA bacterial genes in *S. grammicus* embryos and their amniotic environment and compared it with the diversity observed in the maternal intestine, mouth, and cloaca. It is important to note, however, that the presence of bacterial DNA does not necessarily indicate the existence of an active and growing bacterial community, or microbiota colonizing the embryonic environment. Nonetheless, transient biological interactions may occur between microorganisms and the developing embryo (1). Bacteria can induce components of conserved pathways in early animal development (34). However, the possibility of bacterial DNA crossing over to the embryonic environment is low. Studies conducted on mice (35) and goats (36) have shown that the placental barrier provides effective protection against DNA. In mice, however, fetal DNA can be detected in maternal blood in small amounts, similar to the presence of cell-free fetal DNA in maternal plasma observed in humans (37).

We found a significant, but partial, overlap between the embryonic and maternal bacterial microbiota. For instance, amplicon sequence variants were detected in the embryonic GI tract but not in maternal tissues. These bacteria are likely to enter the embryos during the early stage of development, as the maternal bacterial microbiota changes during gestation (38). It is possible, therefore, that the maternal microbiota begins to be molded for the vertical transfer of specific bacteria to its offspring. The bacterial seeding from maternal sources might be a continuous rather than a one-time event, and the early colonization process is a balance between bacterial entry and selection (39). To confirm this, similar research can be done with lizards in the early stages of gestation. This balance is likely to play a crucial role in the lizard's developmental program, possibly leading to a more limited diversity of bacterial ASVs in the

embryos compared to the maternal tissues. Such limited diversity was observed in this study and has been observed in other vertebrates (40). Once born, animals are exposed to a large variety of environmental bacteria from their habitat, consumed diet, social relationships, etc. (41, 42). While the placental environment is considered a low biomass environment with limited diversity (40), the microorganisms in the placental environment can significantly affect embryonic development. Numerous examples of pathogenic bacteria invading the placental environment with detrimental effects are well documented (43–46).

Nearly all ASVs found in the membrane and yolk were present in maternal tissues, whereas the amniotic fluid had an intermediate level. It is possible that the membrane and yolk act as filters for microbial seeding, and the transient microorganisms seen in the amniotic fluid are on their way to the embryo. The model organism, the Hawaiian bobtail squid (*Euprymna scolopes*) establishes a mutualistic interaction with the Gram-negative bacteria *Vibrio fischeri*. During embryogenesis, the squid produces a complex cocktail of antimicrobial peptides (AMPs) to prevent the colonization of the hatchling organ surface by Gram-positive bacteria (47, 48). A similar composition of ASVs among embryos from different mothers was observed, which may suggest the existence of strong transmission controls such as mechanisms for bacterial recruitment and selection (49). The most abundant ASVs in the embryos of *S. grammicus* were mostly Gram-negative bacteria (see Fig. 3). So, one hypothesis might be that during the formation of the extraembryonic membranes and yolk, these structures produced AMPs that filter or select specific bacteria.

In placental organisms, such as humans, transplacental trafficking has been proposed as a route of cell exchange between mother and fetus. This mechanism could be involved in the selection of bacteria in the maternal placenta and in the fetal membranes of the embryo. The extraembryonic membrane is closely linked to the *S. grammicus* placenta and contains trophoblasts and specialized cells similar to those present in the human placenta (29, 50). These epithelial cells participate in bacterial selection and/or pathogen defense via AMPs, defensins, and Toll-like receptors (51–53). As such, a second hypothesis, which does not exclude the previous one, is that given the placental function as the interface between the maternal and fetal environment, it is possible that the placenta constitutes the first filter between the maternal and fetal environment. Nonetheless, the exact mechanisms governing the restriction and/or promotion of microbial access remain undefined (51).

In mammals, early exposure to the peptidoglycan (PG) from Gram-negative commensals is especially important for proper gastrointestinal-associated lymphoid tissue maturation (54). The lack of exposure to bacterial symbionts or exposure to pathogens during this critical period can have negative effects on the health of the GI tract and the overall immune response (55). Both Gram-positive and Gram-negative bacteria, whether free-living or associated with a host, release PG fragments into the surrounding environment when their cell walls are damaged or restructured. The host detects these PG fragments as a sign that bacteria are present. However, in some commensal or mutualistic Gram-negative species, significant amounts of the PG monomer are released, which acts as a signal mediating a specific response in the epithelium of susceptible hosts (47, 56).

The bacterial microbiota in embryonic tissues had the greatest overlap (similarity) with those found in the maternal mouth and aseptic ventral skin, followed by those in the cloaca. The mucosa of the cloaca and mouth of reptiles share some similarities, such as being moist, slimy layers that line the inner surfaces of their respective structures. They both contain glandular cells that produce mucus and other substances to maintain moisture and protect the mucosal surface (57–59). However, their structure and function differ significantly, with the mucosa of the mouth being related to feeding, chewing, and digestion, while the mucosa of the cloaca is related to the elimination of waste and reproduction. Although there are some environmental similarities and some bacterial ASVs were shared between these two structures in this study, it is likely

that they host different types of bacteria due to their physiological distinct functions (60). The bacteria found in the aseptic ventral skin were the bacteria most intimately interacting with the epithelium, as the aseptic method probably eliminated the more superficial layers of bacteria. Although we identified many shared ASVs between the maternal mouth and aseptic ventral skin with the embryonic tissues, it is challenging to conclusively determine that these ASVs originate from these maternal tissues. Individuals often harbor similar bacterial species, so identifying transmission based solely on species-level matches is insufficient. Bacterial species can include multiple variants of subspecies strains that are specific to different individuals (39). One possible alternative is to use strain-level typing profiles to detect and quantify cases of maternal-embryo transmission.

It is known that human babies delivered by C-section harbor bacteria in their intestines similar to those found on the maternal skin (8). While the exact mechanism of bacterial transfer from maternal skin to the newborn's intestines is still debated, the prevailing hypothesis suggests that skin bacteria are transmitted during skin-to-skin interactions between the newborn and the mother or medical personnel (61). In our study, we observed a high similarity between the bacteria present on the maternal skin and those found in the embryos, despite the embryos never coming into direct contact with skin tissues. Our findings indicate that skin bacteria may potentially be transmitted before birth and influence vertebrate development.

The high similarity of maternal oral bacteria with bacteria in the embryo has also been detected in human maternal transmission studies (40, 62). The role and biological implication of these oral commensal bacteria in maternal transmission are still unknown. One proposed explanation is that oral bacteria facilitate hematogenous transmission by binding to the vascular endothelium and altering its permeability, thus functioning as "enabler" so that other commensal bacteria can reach the placental environment (40). This ability to destroy the integrity of the endothelium was attributed to *Fusobacterium nucleatum* (63). This bacterium commonly identified in the mouth contains adhesins that bind to vascular endothelial cadherin.

In this study, a low level of overlap between adult and embryonic intestinal ASVs was detected. Kohl et al. (14) reported that the fecal bacteria of neonate lizards from the viviparous species *Phymaturus williamsi* overlapped 34% ± 6% with that of their mothers, while in this study, we found a lower level of overlap (15% ± 10%) between adult *S. grammicus* and embryonic intestinal ASVs. In newborn humans, several bacterial species from the vaginal or skin bacterial microbiota can be detected in the first days of life, but this is transient and they are eventually replaced by the typical intestinal bacteria (39). This could be due to substantial differences in environmental conditions between the placenta and the GI tract after birth, such as the type and form of nutrients ingested by the newborn, which may not favor the survival of skin bacteria in the intestinal environment. For instance, when solid plant-derived food is given to human infants, bacterial communities involved in the metabolism of plant polysaccharides can be found in their gut (64, 65). In adult *S. grammicus* lizards, bacterial communities also showed a low overlap between the small intestine and the mouth (4.7% ± 2.4%), aseptic ventral skin (4.8% ± 1.6%), and cloaca (5.1% ± 2.3%).

## Bacterial groups and putative functions in the maternal and embryonic samples

The maternal samples were found to be dominated by bacteria from the phyla Proteobacteria and Firmicutes. In humans, Proteobacteria was the most abundant phylum in both the amniotic fluid and placenta (66) and also found in female pandas toward the end of pregnancy (67). *Lachnoclostridium* and Lachnospiraceae were more abundant in the maternal small intestine of *S. grammicus*. The Lachnospiraceae family is known to be a part of the intestinal microbiota in animals and is involved in the fermentation of complex carbohydrates (62, 68). *Lachnoclostridium* produces short-chain fatty acids, such

as butyrate, which contribute to the permeability of human embryo umbilical cords (69, 70).

Bacterial groups, such as Enterobacteriaceae and *Hafnia,* were identified in the maternal cloaca. Previous studies have shown that they are enriched in the GI tract and in the reproductive system of overweight pregnant women (71, 72). These studies reported that these taxa induced maternal insulin insensitivity and adiposity, which facilitate energy storage and increased nutrient availability in the plasma for transfer to the embryo (71). It is noteworthy that *Helicobacter* showed a significantly high relative abundance in the mouth and cloacal samples, particularly in the cloaca, compared to the other maternal samples. A similar high prevalence of *Helicobacter* has been observed in oral and cloacal samples of Orinoco crocodiles (73), raising concerns about potential zoonotic disease transmission. Further research, however, is needed to investigate in more detail the possible function of *Helicobacter* as a commensal bacterial microbiota in reptiles. Nonetheless, to better understand the role of these maternal bacterial taxonomic groups in the embryonic development in lizards, it would be important to examine bacterial changes during gestation and correlate them with those found in the embryo and amniotic environment. Additionally, studying the changes in the microbiota and metabolic profiling in the mothers during gestation could provide insights into their possible roles in embryonic development.

*Curvibacter lanceolatus, Stenotrophomonas,* and *Brevundimonas* were the most abundant genera in the embryonic tissues, maternal mouth, cloaca, and aseptic ventral skin of *S. grammicus. Stenotrophomonas* has been detected in cerebrospinal fluid and blood of humans (74). Their primary functions are reported to be the degradation of carbohydrates (75, 76). It is notable that *Stenotrophomonas* is also abundant in the reptilian embryonic tissues. This suggests that *Stenotrophomonas* may play an important role in these environments, although further research is needed to understand the specific functions of this bacteria.

The ASVs belonging to *Ralstonia* were exclusive of the embryonic GI tract. It is difficult to speculate why members of *Ralstonia* were found only in the embryonic GI tract of the embryos. These bacteria can colonize different parts of the animal's body, including the oral cavity (77), GI tract (78), reproductive organs (79), and have been found in *S. grammicus'* feces (80). *Ralstonia* has been reported to be part of the central microbiota of human breast milk and contributes to the development and maturation of the immune system (81, 82). In humans, *Ralstonia* has been identified as a resident of the maternal-embryonic interface, such as in the endometrial decidua, placenta, chorion, and amnion (83, 84).

We investigated the potential functions of bacterial communities in both the embryo and the mother to understand their possible roles. Predicted functions related to the degradation of mucin, such as fucose and rhamnose degradation pathways, were more abundant in the maternal samples (mouth, cloaca, and small intestine). This suggests the presence of bacteria that utilize mucin as a carbon source (85). Additionally, functions related to carbohydrate metabolism indicate the presence of strictly anaerobic bacteria in the maternal intestine and aerobic or facultative anaerobic bacteria in the embryonic GI tract (86). Another putative function that was more abundant in the maternal samples was related to the biosynthesis of vitamin K, specifically "the biosynthesis superpathway of menaquinol and demethylmenaquinol." Vitamin K is a secondary metabolite produced by mutualistic bacteria, highlighting the association of the bacterial microbiota with its host and showing the more complex bacterial community in the mother compared to that in the embryo. Additionally, pyruvate and succinate fermentation pathways were more abundant in maternal samples than in embryonic samples. These pathways produce short-chain fatty acids, such as acetate, propionate, and butyrate (87). These metabolites can be used by the maternal intestinal epithelial cells as an energy source and may contribute to cell differentiation of the embryonic epithelium (88, 89).

Pathways related to the biosynthesis of cofactors and vitamins, such as NAD and heme group biosynthesis, were abundant in embryonic samples. These molecules are

acceptors or donors of electrons and recover energy metabolism and participate in glycolysis, gluconeogenesis, and oxidative stress (90). These pathways might play an important role in obtaining energy for the developing embryo and generate a redox balance in the amniotic environment.

A function related with metabolism L-leucine was very abundant in the embryonic and maternal bacterial communities. Leucine has been identified as an important precursor for the biosynthesis of fatty acids in embryonic tissues (91). Fatty acids and proteins are significant constituents of embryonic tissues mainly in the yolk (92). During development, the mother supplies precursors for these components to the embryo through the fetal membranes, which could explain the high abundance of this function in both the mother and the embryo. The CMP-legionaminate biosynthesis was exclusive in the embryonic and maternal intestine. This function is implicated in the immune system modulation (93). The bacterial processes involved in the health and development of the host occur in the small intestine, e.g., the maturation of the immune system. This might indicate that the bacteria-intestine interaction is participating in the maturation of the immune system of the developing embryos of the lizard *S. grammicus*.

## Ecological and evolutionary impacts of vertical transmission of microbiota in reptiles

The viviparous lizard *S. grammicus* has developed adaptations to harsh environmental conditions in the high mountains of central Mexico. These adaptations include changes in morphology, physiology, behavior, intestinal microbiome, and life history (80, 94–97). Evidence from other *Sceloporus* species indicates that adaptations in the placenta may allow for the exchange of nutrients to keep embryos developing under restrictive environmental conditions (98). There is evidence of uterine transfer and secretion, as well as fetal absorption in other viviparous lecithotrophic species (99), potentially affecting important life history events (50, 100). Bruijning et al. (49) demonstrated that vertical transmission of microorganisms during embryonic development can increase phenotypic variation between hosts, potentially playing a role in adaptive radiation in lizards. This phenomenon occurred when a single ancestral lizard species evolved into multiple distinct species, each adapted to different ecological niches. The ability of lizards to adapt to different environmental conditions, combined with selective pressure from various ecological niches, may have contributed to adaptive radiation in lizards (101–103). Our findings suggest the possibility that the viviparous lizard *S. grammicus* has developed adaptations to harsh environmental conditions, potentially involving adaptations in the placenta, including mechanisms of vertical microbiota transmission during embryonic development. Previous studies found that lecithotrophic reptiles rely on the yolk nutrients for embryo development, while calcium and small-sized nutrients are provided through the placenta (104). The results reported in this study showed that a selective group of bacteria found in the embryo are also obtained through the placenta. The function of the placenta to incorporate or restrict the inclusion of these bacteria remains to be found.

Our findings also highlight the significance of vertical transmission of microorganisms during embryonic development as a crucial maternal effect that can affect the adaptation of offspring to their surroundings. Maternal effects refer to the influence that a mother has on the phenotypic traits of her offspring, regardless of gene inheritance (105). This can happen through diverse mechanisms, such as nutrient, hormone and the transfer of other molecules through the placenta or yolk, mother-offspring interaction during parental care, and exposure of the offspring to her microbiota. Whether the latter has an adaptive value for the mother or for the embryo is still unknown. However, some proposed mechanisms suggest that the fetus can obtain energy from the mother's blood more efficiently or that metabolite-producing bacteria, e.g., butyrate, maintain the functions of the fetal gastrointestinal epithelium and promote immunological tolerance of the mother (72, 106).

## Conclusion

In summary, we detected bacterial 16S rRNA genes in the embryos of *S. grammicus*, although the diversity was low and the community structures were similar, which suggest that there are strong controls on the maternal transmission of microorganisms to the embryo. We found a significant but partial overlap between the maternal and embryonic microbiota, suggesting that the transmission of bacteria from the mother to the offspring is a continuous process, and some bacteria may have been transferred during early embryonic stages. The embryonic bacteria were found to overlap mostly with those found in the mouth and aseptic ventral skin, although it is difficult to conclude that the shared ASVs originated from these maternal tissues. Overall, our results highlight the significance of vertical transmission of bacteria during embryonic development in *S. grammicus*.

## MATERIALS AND METHODS

### Study model and site of sampling

*Sceloporus grammicus* is a viviparous and widely studied lecithotrophic lizard, i.e., the mother supplies nutrients to the embryo through the egg yolk. The placenta of *S. grammicus* consists of different extraembryonic membranes that are in contact with the egg and the uterine wall (29). It permits gas exchange, although water, and organic and inorganic molecules can also pass through it (100).

The sampling site was located in Ixtenco, Tlaxcala, Mexico, at 2,600 m above sea level (masl) (19° 14′ N, 97° 55′ W). This area is characterized by a temperate-semi-arid climate with a mean temperature of 14.5 ± 6.6°C and a mean relative humidity of 58.2% ± 28.8%. The vegetation in the area was maize crops (*Zea mays* L.). At 2,600 masl, *S. grammicus* feeds on more than 25 families of Insecta and Arachnida and has an average body size of 10 ± 2 cm (80).

Eight pregnant females with their respective embryos at stage of development 40 (the embryo presents characteristics of a neonate, this includes coloration and differentiated scales) (29) were collected on 25 March 2021. In the field, pregnancy was determined by palpation. The lizards were collected manually, stored, and transported individually in aseptic cloth bags to the "*Laboratorio de Interacciones Bióticas*" of the "*Universidad Autónoma de Tlaxcala*," Tlaxcala. In the laboratory, the lizards were kept individually at 20–25°C in the dark to avoid stress until their euthanasia and dissection the day after they were captured. Each lizard was manipulated with sterile gloves before dissection, i.e., there was never contact between the human and the lizard skin.

### Dissection of the maternal and fetal gastrointestinal tract

All downstream experimental procedures were done under strict sterile conditions in a Class II Biosafety Cabinet (NuAire, MN, USA). Before dissection, mouth and cloaca samples were taken by scraping with rayon swabs (COPEN, Italy). Swabs were immediately placed in sterile polypropylene 1.5 mL tubes and stored at −20°C. After swabbing, individuals were placed on a sterile dissecting plate and disinfected using sterile gauze pads impregnated first with a generous amount of hydrogen peroxide 20% (vol/vol), second with sodium hypochlorite 2% (vol/vol), third with ethanol 70% (vol/vol), and then rinsed twice with sterile water. The material used for disinfection was discarded to avoid contamination by scales or any waste from the lizard. Once the individual was disinfected, all manipulations were done with new and sterile dissecting plates and sterile nitrile gloves (UniSeal, China). They were changed frequently to avoid cross-contamination. Subsequently, a swab sample from the ventral body region was taken as a control of our aseptic technique. Using sterile instruments, the ventral skin of the lizard was cut to expose the internal organs. The external skin was never touched by the dissection instruments or other organs. Subsequently, the intestinal tract was carefully removed to reveal the embryos *in utero*. Surgery instruments were cleaned and flamed

every time between surgical cuts. The embryos were extracted from the maternal body, placed individually in sterile Petri dishes, and kept at 4°C until dissected for their GI tract immediately after the adult dissection. The maternal GI tract was extracted, and the small intestine (the region between the stomach and the rectum, just before the cecum) was dissected and stored in sterile tubes at −20°C until DNA extraction the next day.

## Dissection of the embryos

Approximately, two embryos were analyzed per mother ($n = 17$). First, approximately 100–200 µL of the amniotic fluid was collected with insulin syringes (BD Ultra-Fine, 31G × 8 mm). The membrane was removed and placed in a tube, and the yolk, approximately 50–100 µL collected with insulin syringes. The embryo was transferred to a sterile dissecting plate and dissected to sample the GI tract. Each embryo was dissected with new and sterile dissection plates, gloves, and sterile surgical instruments. All embryonic samples were placed and stored in sterile propylene tubes at −20°C until their subsequent DNA extraction within 5 days. Only samples that were free from cross-contamination, such as membrane with yolk, were considered for analysis. As a result, we were able to retrieve 6 membrane samples, 4 yolk samples, and 6 amniotic fluid samples out of a total of 17 embryos.

## DNA extraction, 16S rRNA gene amplification, and sequencing

The extraction of DNA from the fluids (amniotic fluid and yolk), tissues (maternal and embryonic GI tract, and membrane), and swabs (mouth, cloaca, and aseptic ventral skin) was done with the UCP Pathogen Mini kit (Qiagen, Germany). The isolation kit was selected to increase microbial DNA extraction over DNA from the host. The obtained DNA was stored at −20°C for library construction. Blank controls were included in each extraction batch. Blank controls were verified for contamination by electrophoresis and amplification of the 16S rRNA gene. The construction of libraries was done through the amplification of the V3-V4 regions of the 16S rRNA gene with the 341F (5′-CCTACGGGNGGCWGCAG-3′) and 805R (5′-GACTACHVGGGTATCTAATCC-3′) primers. All primers were modified with the sequencing platform adapter and an 8-nt tag. The conditions of the PCR are described in reference (80). The amplification of the 16S rRNA genes was done five times, and the products were visualized on a 1% agarose gel. The amplification products were combined in a single mixture and purified with the DNA Clean Kit (Zymo research, Irvine, CA, USA). The purified amplicons were quantified with the Quant-iT PicoGreen dsDNA Assay Kit (Invitrogen, Waltham, MA) on a Nanodrop 3300 Fluorospectrometer (Thermo Fisher Scientific, Waltham, MA), normalized, and combined in equimolar amounts for sequencing. Blank, positive, and reagent controls were included in each PCR run. All these controls were included in the amplification process with the sequencing adapters, but no amplicons were obtained. Sequencing was done by Macrogen Inc (Seoul, South Korea) using the Illumina MiSeq platform and 300 bp PE runs (Illumina, San Diego, CA, USA). The raw sequences were analyzed to identify human DNA sequences, which could serve as potential indicators of contamination during the sequencing process. No human DNA was detected.

After sequencing, the following samples contained sufficient reads for downstream analysis: maternal small intestine ($n = 5$), mouth ($n = 8$), cloaca ($n = 8$) and aseptic ventral skin ($n = 8$), and embryonic GI tract ($n = 17$), amniotic fluid ($n = 9$), membrane ($n = 6$), and yolk ($n = 4$) (Fig. 1).

## Bioinformatic and statistical analysis

The sequencing data were processed with the Quantitative Insights Into Microbial Ecology 2 (QIIME 2-2021.4) software (107). The sequences were demultiplexed, the noise of the sequencing was eliminated, paired-end reads were joined, low-quality reads, singletons, and chimeric sequences were eliminated, and the ASVs were inferred using DADA2 (108) with the following parameters: --p-trunc-len-f 250, --p-trunc-len-r

200, --p-max-ee-f 2, --p-max-ee-r 2, and --p-pooling-method "pseudo." The taxonomic assignment of the ASVs was done with the Sklearn classifier based on the Naive Bayes algorithm using the SILVA version 138 reference database and trained in the V3-V4 regions of gene 16S rRNA (109). An alignment-based filtering method was applied to the representative sequences using vsearch with the following parameters: perc-identity = 0.97 and perc-query-aligned =0.95. Sequences that did not align were removed. Host DNA and organellar 16S rRNA sequences, i.e., from mitochondria and chloroplast, were deleted. The *decontam* R package 1.14.0 was used to remove reagent contaminants post-sequencing (31). This analysis is recommended in studies of low biomass samples, leading to outline microbiota with a greater possibility of representing the real community. The functions of the bacterial communities were predicted with PICRUSt 2.0.0 (110). The data were aligned with the reference genome database of Integrated Microbial Genomes with Markov models. The abundance of metabolic routes was based on enzyme classification numbers (EC numbers) and was inferred with MetaCyc (111).

All statistical analyses were performed with R version 4.1.2 (112). Alpha taxonomic diversity was calculated with the ASV observation table and determined with Hill numbers at diversity orders $q = 0, 1$, and 2 with the MetagenomeDiversity R script (113). Significant differences in the effective number of ASVs between maternal and embryonic samples, and across sections were tested with linear mixed effect models using the individual identity (maternal or embryonic accordingly) as a random factor with the "lme" function in *nlme* R package 3.1-162 (114). The significance of the effect was calculated with 1,000 Monte Carlo simulations using the *pgirmess* 2.0.0 package in R (https://github.com/pgiraudoux/pgirmess).

To select the ASVs and predicted functions that most contributed to the differences in compositions between samples to plot in the heatmaps, we applied a supervised learning method using the *q2-sample-classifier* 2023.5.0.dev0 plugin within QIIME 2 (115). We selected the ASVs and predicted functions with the top 50 highest score of importance (importance of each input feature to model accuracy). The table of frequencies of ASVs and predicted functions was used to calculate the relative abundance of bacterial taxonomic groups and functions of embryonic and maternal samples. The heatmaps were constructed using the *ComplexHeatmap* R package 2.13.2 (116).

A compositional approach was used to investigate the structure and composition of bacterial communities (117). First, the zero count values were replaced on the observations table of ASVs with a count zero multiplicative method using the "cmultRepl" function in *zCompositions* R package 1.4.0-1. Data sets were centered log-ratio (*clr*) transformed with the "codaSeq.clr" function in *CoDaSeq* R package 1.3.4 (118). The maternal and embryonic bacterial structures were explored by PCA using the *clr* transformed table. The significance differences between maternal and embryonic bacterial communities were tested with perMANOVA using the adonis function in the *vegan* R package 2.6-4 (119) using Aitchison distances (118). Pairwise comparisons of Aitchison distances between the same individual and between the embryonic samples with samples from its mother were manually selected to test the difference in the distances among maternal and embryonic samples. Significant differences were determined with Fisher-Pitman permutation test with 10,000 Monte-Carlo permutations using *coin* R package 1.4-2 (120).

We used the method described in reference (33), based on the Jaccard dissimilarity index, to determine the partition of beta diversity into two components, turnover and nestedness. We used the equation:

$$\beta_{CC} = \beta_{-3} + \beta_{rich} \qquad \text{(Eq. 1)}$$

where: total dissimilarity ($\beta_{CC}$), turnover ($\beta_{-3}$), and nestedness by richness difference ($\beta_{rich}$).

The Venn diagrams were constructed with shared species of the embryonic and maternal samples with the "get_set" function and VennDiagram R 1.7.3 package (121). The matrix of shared ASVs was obtained with the function *betapart.core* of the *betapart*

R package 1.5.6 (122). The overlap was calculated relative to the ASVs in the embryonic tissues as the percentage of the embryonic ASVs shared with the maternal ASVs.

FEAST analysis (123) was done to determine the contribution of the different maternal sources to the embryos. The analysis was done with each of the mothers and their respective embryos. Maternal samples were established as sources and embryonic samples as sinks. All the scripts are available at Github (https://github.com/NinaMon-toya/Article-Transfer).

## ACKNOWLEDGMENTS

The authors thank "Estación Científica La Malinche," Erick Gómez, Miguel Domínguez, and Mauricio Hernández for their logistic support during fieldwork, Mario Hernández for his help in the bioinformatics analysis, and Microbioma Lab for access to computing resources.

This research was funded by "Consejo Nacional de Ciencia y Tecnología" (CONACyT), Ciencia de Frontera project number: CF-2019–137748, Infraestructura project number INFRA-205945 and Cátedras CONACyT program (project number: 883). N.M.-C. received a PhD grant-aided support (scholarship number 703251), and S.H.-P. (grant number: 929602) and E.S.G.-A. a postdoctoral grant from CONACyT.

Y.E.N.-N. and A.H.D.V.-P. conceived the original idea; A.H.D.V.-P. performed the fieldwork; N.M.-C., L.D., and Y.E.N.-N. performed the molecular biology analysis, S.H.-P., N.M.-C., E.S.G.-A., and Y.E.N.-N. analyzed the data; N.M.-C. and Y.E.N.-N. interpreted the results and wrote the first draft of the manuscript. All the authors revised and approved the final version of the manuscript.

## AUTHOR AFFILIATIONS

[1]Doctorado en Ciencias Biológicas, Centro Tlaxcala de Biología de la Conducta, Universidad Autónoma de Tlaxcala, Tlaxcala, Mexico
[2]Estación Científica La Malinche, Centro Tlaxcala de Biología de la Conducta, Universidad Autónoma de Tlaxcala, Tlaxcala, Mexico
[3]Laboratory of Soil Ecology, CINVESTAV, Mexico City, Mexico
[4]Departamento de Zoología, Instituto de Biología, Universidad Nacional Autónoma de México, Ciudad de México, Mexico
[5]Laboratorio de Interacciones Bióticas, Centro de Investigación en Ciencias Biológicas, Universidad Autónoma de Tlaxcala, Tlaxcala, Mexico
[6]Consejo Nacional de Ciencia, Humanidades y Tecnología-Centro Tlaxcala de Biología de la Conducta, Universidad Autónoma de Tlaxcala., Tlaxcala, Mexico

## AUTHOR ORCIDs

Stephanie Hereira-Pacheco http://orcid.org/0000-0003-1433-8187
Luc Dendooven http://orcid.org/0000-0002-4148-2283
Aníbal H. Díaz de la Vega-Pérez http://orcid.org/0000-0001-5651-4383
Yendi E. Navarro-Noya http://orcid.org/0000-0002-7191-7815

## FUNDING

| Funder | Grant(s) | Author(s) |
| --- | --- | --- |
| Consejo Nacional de Ciencia y Tecnología (CONACYT) | 137748 | Yendi E. Navarro-Noya |
| Consejo Nacional de Ciencia y Tecnología (CONACYT) | 703251 | Nina Montoya-Ciriaco |
| Consejo Nacional de Ciencia y Tecnología (CONACYT) | 929602 | Stephanie Hereira-Pacheco |

## AUTHOR CONTRIBUTIONS

Nina Montoya-Ciriaco, Data curation, Formal analysis, Investigation, Methodology, Visualization, Writing – original draft, Writing – review and editing | Stephanie Hereira-Pacheco, Data curation, Formal analysis, Writing – review and editing | Arturo Estrada-Torres, Supervision, Writing – review and editing | Luc Dendooven, Funding acquisition, Project administration, Resources, Supervision, Writing – review and editing | Fausto R. Méndez de la Cruz, Supervision, Writing – review and editing | Elizabeth Selene Gómez-Acata, Formal analysis, Supervision, Writing – review and editing | Aníbal H. Díaz de la Vega-Pérez, Conceptualization, Supervision, Writing – review and editing | Yendi E. Navarro-Noya, Conceptualization, Formal analysis, Funding acquisition, Investigation, Methodology, Project administration, Resources, Supervision, Validation, Visualization, Writing – original draft, Writing – review and editing

## DATA AVAILABILITY

All raw sequences are available in the NCBI Sequence Read Archive (SRA) with the BioProject accession number PRJNA963006.

## ETHICAL APPROVAL

The collection of lizards was done with the permission of the Secretary of Environment and Natural Resources (SEMARNAT) (official letter SGPA/DGVS/00107/20). The lizards were manipulated and euthanized following the Official Mexican Standard NOM-062-ZOO-1999.

## ADDITIONAL FILES

The following material is available online.

### Supplemental Material

**Supplemental material (Spectrum01780-23-s0001.docx).** Tables S1 and S2; Fig. S1 and S2.

### Open Peer Review

**PEER REVIEW HISTORY (review-history.pdf).** An accounting of the reviewer comments and feedback.

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
