## [Reviewer comments · Microbiology Spectrum]

Microbiology Spectrum

Maternal transmission of bacterial microbiota during embryonic development in a viviparous lizard

Nina Montoya-Ciriaco, Stephanie Hereira-Pacheco, Arturo Estrada-Torres, Luc Dendooven, Fausto R. Méndez de la Cruz, Elizabeth Gómez-Acata, Anibal Díaz de la Vega-Pérez, and Yendi Navarro-Noya

Corresponding Author(s): Yendi Navarro-Noya, Universidad Autónoma de Tlaxcala

Review Timeline:

Submission Date:	May 9, 2023
Editorial Decision:	June 19, 2023
Revision Received:	August 22, 2023
Accepted:	September 8, 2023

Editor: Diyan Li

Reviewer(s): Disclosure of reviewer identity is with reference to reviewer comments included in decision letter(s). The following individuals involved in review of your submission have agreed to reveal their identity: Avvari Bhaskara Balaji (Reviewer #1)

Transaction Report:

DOI: <https://doi.org/10.1128/spectrum.01780-23>

June 19, 2023

Dr. Yendi E Navarro-Noya
Universidad Autónoma de Tlaxcala
Centro de Investigación en Ciencias Biológicas
Tlaxcala, Tlaxcala 90000
Mexico

Re: Spectrum01780-23 (**Maternal transmission of bacterial microbiota during embryonic development in a viviparous lizard**)

Dear Dr. Yendi E Navarro-Noya:

Please note that the reviewer's comment that the major concern in the field is sample contamination and given the reduced diversity in the samples match the controls, this does bring up the fact that certain bacteria may persist more than others and may be overrepresented.

Link Not Available

Sincerely,

Diyang Li

Journals Department
Reviewer comments:

Reviewer #1 (Comments for the Author):

Dear Authors,

Clarity and Organization:

- The description of the samples analyzed and the sequencing methods used is clear and well-organized.
- However, consider providing more specific information on the number of samples analyzed for each category (maternal vs.

embryonic) to better understand the statistical analysis.

Alpha Diversity:

- The use of Hill numbers to analyze alpha diversity is appropriate.
- It is interesting to note that maternal samples had significantly higher diversity than embryonic samples, but further explanation or discussion on the possible reasons behind this difference would be valuable.

Taxonomic Composition:

- The description of the taxonomic composition of the embryonic and maternal samples is informative.
- Discuss any notable differences in taxonomic composition between different sample sites within the maternal samples.

Community Composition and Structure:

- The identification of specific amplicon sequence variants (ASVs) in both maternal and embryonic samples adds depth to the analysis.
- Provide more context or discussion on the potential roles or functions of the identified ASVs in the microbiota.
- The principal component analysis (PCA) and permutational multivariate analysis of variance (perMANOVA) results are mentioned briefly. Consider expanding on the interpretation of these results and their implications.

Reagent Contamination:

- The consideration of potential reagent contamination and the negative results in the reagent controls is commendable.
- Mention the steps taken to minimize or control for contamination during the sample collection and sequencing processes.

Functional Prediction:

- The use of PICRUSt analysis to predict the potential functions of the bacterial microbiota is appropriate.
- Elaborate on the functional implications of the observed differences in potential functions between embryonic and maternal tissues.
- Discuss any functional patterns or pathways that are unique to specific sample sites or stages of development.

Overlap between Embryonic and Maternal Microbiotas:

- The comparison of ASVs between embryonic and maternal samples provides valuable insights into the potential transfer of microbiota.
- Discuss the biological significance of the overlapping ASVs and the implications for vertical transmission of the microbiota.

Overall, the "Results" section provides detailed information on the taxonomic and functional composition of embryonic and maternal microbiotas. However, it would be beneficial to expand on the interpretation and biological relevance of the findings, consider discussion of vertical transmission contamination on human's motherhood to the fetus is often by physical rupture or through blood circulation.

Reviewer #2 (Comments for the Author):

Regarding the microbial transmission from mother to child, due to sampling difficulties and experimental contamination problems, no more definite conclusions have been drawn so far. In this study, this study innovatively used viviparous lizard as the study subject and observed the maternal microbial transmission during embryonic development.

There are two minor issues:

1. In the analysis of maternal microbial transmission, the "betapart" method was used, while in other microbial traceability studies, methods such as "Sourcetracker" or "FEAST" are often used. What is the basis for the authors' choice of the former? What is the difference between the former and the latter? Which method is superior?
- 2, the horizontal coordinates of Figure 7, the different color bars, I guess should be "the amniotic fluid, embryonic gastrointestinal (GI) tract, membrane and yolk of 40 stage embryos", these should be specifically marked out

Reviewer #3 (Comments for the Author):

In this study, Montoya-Ciriaco et al, study the transmission of maternal microbiota to embryos. They use high throughput sequencing of the 16s rRNA bacterial genes to evaluate the relative composition of embryo and amniotic fluid bacteria compared to maternal tissues. The authors find reduced diversity in the bacterial composition in embryos compared to maternal tissue suggesting a controlled vertical microbiota transmission. The authors propose that maternal microbiota was transmitted, early in embryo development, mainly from the mouth and aseptic ventral skin to the embryos based on amplicon sequence variants. The field is still debating whether maternal-fetal microbiota transfer occurs early in embryo although emerging lines of evidence appear to support this hypothesis. A lingering concern in the field has been the possibility of low-level contamination in the samples. The authors try to address this concern in the paper. Overall, the study is straightforward and the reduced diversity in the bacterial population in embryos matches reports from human newborn meconium. The data supports the hypothesis, although more work would be needed to definitely point to the importance and role of the microbiota in reptile development, which the authors refer to but do not address in this study. In addition, where and how the selection/filtration of bacterial species occur still remains elusive. I have a few comments that are detailed below:

It seems that the bacteria that are relatively low abundance in the mother are being transmitted disproportionately to the embryo. It also appears to be higher than other maternal tissues in the control. The authors mention the relative abundance in figure 2 but how different is the total amount of bacteria in the different samples? Can this be estimated from amplification

amplitude? Can the authors comment on this?

It is also curious that the pattern of the 50 most abundant ASVs is most similar to the mouth and the non-superficial bacteria in the control (aseptic ventral skin). That is puzzling. Are the bacteria closer to the epithelium just harder to eliminate and perhaps overrepresented in samples somehow?

The authors mention 3 ways of transferring bacteria to the embryo - placenta, vaginal spread, intestinal absorption. Transplacental trafficking has been suggested widely. The amniotic fluid looks very similar to other embryo tissues, so the maternal control in terms of which bacteria are transmitted, would have occurred upstream. The membrane and the yolk share a lot of ASVs with the mother but the selection has already occurred given the ASVs still belong to a few taxonomic groups. The filtration/selection could happen when passing through the membrane but I find it hard to envision a species-specific filtration system for microbiota in the membrane. How would that work?

Is the diversity in pregnant female lizards different than non-pregnant or nulliparous ones? This would be interesting in light of the fact that proteobacteria tend to increase upon pregnancy. It is possible that there is a pre-defined bias very early in pregnancy that sets the system up for a biased transmission of certain bacteria. This is highlighted by the 28% of ASVs in the embryonic GI tract that are not shared with any maternal sample. The authors mention that these may come from earlier gravid stages and again that comes back to investigating time resolved maternal microbiota distribution during gestation. It is possible that the maternal microbiota is already being selectively shaped for vertical transmission.

A minor comment would be to tone down the language for whether the bacterial transmission is crucial for lizard development and adaptation in the abstract as none of that was tested in this study.

Staff Comments:

Preparing Revision Guidelines

Please return the manuscript within 60 days; if you cannot complete the modification within this time period, please contact me. If you do not wish to modify the manuscript and prefer to submit it to another journal, please notify me of your decision immediately so that the manuscript may be formally withdrawn from consideration by Microbiology Spectrum.

Dear Authors,

Review comments:

Clarity and Organization:

- The description of the samples analyzed and the sequencing methods used is clear and well-organized.
- However, consider providing more specific information on the number of samples analyzed for each category (maternal vs. embryonic) to better understand the statistical analysis.

Alpha Diversity:

- The use of Hill numbers to analyze alpha diversity is appropriate.
- It is interesting to note that maternal samples had significantly higher diversity than embryonic samples, but further explanation or discussion on the possible reasons behind this difference would be valuable.

Taxonomic Composition:

- The description of the taxonomic composition of the embryonic and maternal samples is informative.
- Discuss any notable differences in taxonomic composition between different sample sites within the maternal samples.

Community Composition and Structure:

- The identification of specific amplicon sequence variants (ASVs) in both maternal and embryonic samples adds depth to the analysis.
- Provide more context or discussion on the potential roles or functions of the identified ASVs in the microbiota.
- The principal component analysis (PCA) and permutational multivariate analysis of variance (perMANOVA) results are mentioned briefly. Consider expanding on the interpretation of these results and their implications.

Reagent Contamination:

- The consideration of potential reagent contamination and the negative results in the reagent controls is commendable.
- Mention the steps taken to minimize or control for contamination during the sample collection and sequencing processes.

Functional Prediction:

- The use of PICRUSt analysis to predict the potential functions of the bacterial microbiota is appropriate.
- Elaborate on the functional implications of the observed differences in potential functions between embryonic and maternal tissues.
- Discuss any functional patterns or pathways that are unique to specific sample sites or stages of development.

Overlap between Embryonic and Maternal Microbiotas:

- The comparison of ASVs between embryonic and maternal samples provides valuable insights into the potential transfer of microbiota.
- Discuss the biological significance of the overlapping ASVs and the implications for vertical transmission of the microbiota.

Overall, the "Results" section provides detailed information on the taxonomic and functional composition of embryonic and maternal microbiotas. However, it would be beneficial to expand on the interpretation and biological relevance of the findings, consider discussion of vertical transmission contamination on human's motherhood to the foetus is often by physical rupture or through blood circulation.

Response to Reviewers

Reviewer comments:

Reviewer #1 (Comments for the Author):

Dear Authors,

Clarity and Organization:

• *The description of the samples analyzed and the sequencing methods used is clear and well-organized.*

Thank you very much! We appreciate all the comments made by the reviewer that helped us to improve our manuscript.

• *However, consider providing more specific information on the number of samples analyzed for each category (maternal vs. embryonic) to better understand the statistical analysis.*

We included more information related to the number of samples analyzed in the results, and materials and methods sections as suggested by the reviewer.

“Taxonomic diversity of embryonic and maternal bacterial microbiota. A total of 65 samples from eight pregnant females of *S. grammicus* in the last stage of development (29) were analyzed through metabarcoding of the bacterial communities using the V3-V4 regions of the 16S rRNA gene and MiSeq 2×300 runs, i.e., maternal small intestine ($n = 5$), and swabs from the mouth ($n = 8$), cloaca ($n = 8$) and the ventral body region after the aseptic technique ($n = 8$) (see materials and methods; hereafter aseptic ventral skin). In addition, the amniotic fluid ($n = 9$), embryonic gastrointestinal (GI) tract ($n = 17$), amniotic membrane (hereafter membrane; $n = 6$) and extraembryonic yolk (hereafter yolk; $n = 4$) of one or two embryos per mother lizard were analyzed (Fig. 1). Sequencing produced a total of 766,270 reads ($11,794 \pm 10,529$ in average per sample) and 780 amplicon sequence variants (ASVs) (Table S1).”

“Only samples that were free from cross-contamination, such as membrane with yolk, were considered for analysis. As a result, we were able to retrieve six membrane samples, four yolk samples, and six amniotic fluid samples out of a total of 17 embryos.”

“After sequencing, the following samples contained sufficient reads for downstream analysis: maternal small intestine ($n = 5$), mouth ($n = 8$), cloaca ($n = 8$) and aseptic ventral skin ($n = 8$), and embryonic GI tract ($n = 17$), amniotic fluid ($n = 9$), membrane ($n = 6$) and yolk ($n = 4$) (Fig. 1).”

Alpha Diversity:

• *The use of Hill numbers to analyze alpha diversity is appropriate.*

Thank you for your comment.

• *It is interesting to note that maternal samples had significantly higher diversity than embryonic samples, but further explanation or discussion on the possible reasons behind this difference would be valuable.*

We discussed in more detail the differences in bacterial diversity between maternal and embryonic samples as suggested by the reviewer.

“We found a significant, but partial overlap between the embryonic and maternal bacterial microbiota. For instance, amplicon sequence variants were detected in the

embryonic GI tract, but not in maternal tissues. These bacteria are likely to enter the embryos during the early stage of development, as the maternal bacterial microbiota changes during gestation (38). It is possible, therefore, that the maternal microbiota begins to be molded for the vertical transfer of specific bacteria to its offspring. The bacterial seeding from maternal sources might be a continuous rather than a one-time event, and the early colonization process is a balance between bacterial entry and selection (39). To confirm this, similar research can be done with lizards in the early stages of gestation. This balance is likely to play a crucial role in the lizard's developmental program, possibly leading to a more limited diversity of bacterial amplicon sequence variants in the embryos compared to the maternal tissues. Such limited diversity was observed in this study and has been observed in other vertebrates (40). Once born, animals are exposed to a large variety of environmental bacteria from their habitat, consumed diet, social relationships, etc. (41, 42). While the placental environment is considered a low biomass environment with limited diversity (40), the microorganisms in the placental environment can significantly affect embryonic development. Numerous examples of pathogenic bacteria invading the placental environment with detrimental effects are well-documented (43, 46).

Taxonomic Composition:

- *The description of the taxonomic composition of the embryonic and maternal samples is informative.*

Thank you for your comment.

- *Discuss any notable differences in taxonomic composition between different sample sites within the maternal samples.*

We added this information to the discussion section as suggested by the reviewer.

“*Lachnospiraceae* and *Lachnospiraceae* were more abundant in the maternal small intestine of *S. grammicus*. The *Lachnospiraceae* family is known to be a part of the intestinal microbiota in animals and is involved in the fermentation of complex carbohydrates (67, 61). *Lachnospiraceae* produces short-chain fatty acids, such as butyrate, which contributes to the permeability of human embryo umbilical cords (68, 69).

Bacterial groups, such as *Enterobacteriaceae* and *Hafnia*, were identified in the maternal cloaca. Previous studies have shown that they are enriched in the GI tract and in the reproductive system of overweight pregnant women (70, 71). These studies reported that these taxa induced maternal insulin insensitivity and adiposity, which facilitate energy storage and increased nutrient availability in the plasma for transfer to the embryo (70). It is noteworthy that *Helicobacter* showed a significantly high relative abundance in the mouth and cloacal samples, particularly in the cloaca, compared to the other maternal samples. A similar high prevalence of *Helicobacter* has been observed in oral and cloacal samples of Orinoco crocodiles (72), raising concerns about potential zoonotic disease transmission. Further research, however, is needed to investigate in more detail the possible function of *Helicobacter* as a commensal bacterial microbiota in reptiles.”

Community Composition and Structure:

- *The identification of specific amplicon sequence variants (ASVs) in both maternal and embryonic samples adds depth to the analysis.*

Thank you for your comment.

- *Provide more context or discussion on the potential roles or functions of the identified ASVs in the microbiota.*

We added more information as the reviewer recommended.

“*Curvibacter lanceolatus*, *Stenotrophomonas* and *Brevundimonas* were the most abundant genera in the embryonic tissues, maternal mouth, cloaca, and aseptic ventral skin of *S. grammicus*. *Stenotrophomonas* has been detected in cerebrospinal fluid and blood of humans (73). Their primary functions are reported to be the degradation of carbohydrates (74, 75). It is notable that *Stenotrophomonas* is also abundant in the reptilian embryonic tissues with a probable origin in the maternal tissues. This suggests that *Stenotrophomonas* may play an important role in these environments, although further research is needed to understand the specific functions of this bacteria in these environments.

The ASVs belonging to *Ralstonia* were exclusive of the embryonic GI tract. It is difficult to speculate why members of *Ralstonia* were found only of the embryonic GI tract of the embryos. These bacteria can colonize different parts of the animal’s body, including the oral cavity (76), GI tract (77), reproductive organs (78), and have been found in *S. grammicus*’ feces (79). *Ralstonia* has been reported to be part of the central microbiota of human breast milk and contributes to the development and maturation of the immune system (80, 81). In humans, *Ralstonia* has been identified as a resident of the maternal-embryonic interface, such as in the endometrial decidua, placenta, chorion and amnion (82, 83).”

- *The principal component analysis (PCA) and permutational multivariate analysis of variance (perMANOVA) results are mentioned briefly. Consider expanding on the interpretation of these results and their implications.*

We described in more detail the PCA in the results section as the reviewer recommends.

“A principal component analysis (PCA) revealed two major clusters. The embryonic samples formed a cluster on the right very close to each other indicating small variations in the bacterial structure of the embryonic samples, while the maternal samples formed a more dispersed cluster on the left. The cluster of embryonic samples is separated from the aseptic ventral skin, indicating that the bacterial communities of embryos are different from the bacterial communities of the aseptic control (Fig. 4). The Permutational multivariate analysis of variance (perMANOVA) using Aitchison distances showed a significant difference between the structure of embryonic and maternal bacterial communities, specifically between the amniotic environment, embryonic GI tract, and aseptic ventral skin (aseptic control), which explained the 66% of the total variance ($F = 16.11$, $P < 0.001$, $R^2 = 0.66$). The ASV 3, belonging to the genus *Stenotrophomonas*, was enriched in embryonic samples, specifically in the embryonic GI tract. The ASV 23 belonging to the Lachnospiraceae family was enriched in the maternal small intestine, while the ASV 4 (*Helicobacter*) and 16 (*Hafnia*) were enriched in the maternal cloaca (Fig. 4). The Aitchinson distances among embryonic samples were significantly lower than those among maternal samples ($P < 0.001$; Fig. S1).”

Reagent Contamination:

- *The consideration of potential reagent contamination and the negative results in the reagent controls is commendable.*

Thank you for your comment.

- *Mention the steps taken to minimize or control for contamination during the sample collection and sequencing processes.*

We clarified this in the materials and methods section.

“Each lizard was manipulated with sterile gloves before dissection, i.e. there was never contact between the human and the lizard skin.”

“Dissection of the maternal and fetal gastrointestinal tract. All downstream experimental procedures were done under strict sterile conditions in a Class II Biosafety Cabinet (NuAire, MN, USA). Before dissection, mouth and cloaca samples were taken by scraping with rayon swabs (COPEN, Italy). Swabs were immediately placed in sterile polypropylene 1.5 mL tubes and stored at -20°C. After swabbing, individuals were placed on a sterile dissecting plate and disinfected using sterile gauze pads impregnated first with a generous amount of hydrogen peroxide 20% (v/v), second with sodium hypochlorite 2% (v/v), third with ethanol 70% (v/v) and then rinsed twice with sterile water. The material used for disinfection was discarded to avoid contamination by scales or any waste from the lizard. Once the individual was disinfected, all manipulations were done with new and sterile dissecting plates and sterile nitrile gloves (UniSeal, China). They were changed frequently to avoid cross-contamination. Subsequently, a swab sample from the ventral body region was taken as a control of our aseptic technique. Using sterile instruments, the ventral skin of the lizard was cut to expose the internal organs. The external skin was never touched by the dissection instruments or other organs. Subsequently, the intestinal tract was carefully removed to reveal the embryos in utero. Surgery instruments were cleaned and flamed every time between surgical cuts. The embryos were extracted from the maternal body, placed individually in sterile Petri dishes and kept at 4°C until dissected for their GI tract immediately after the adult dissection. The maternal GI tract was extracted, and the small intestine (the region between the stomach and the rectum, just before the cecum) was dissected and stored in sterile tubes at -20°C until DNA extraction the next day.

Dissection of the embryos. Approximately, two embryos were analyzed per mother ($n = 17$). First, approximately 100-200 μL of the amniotic fluid was collected with insulin syringes (BD Ultra-Fine™, 31GX8mm). The membrane was removed and placed in a tube, and the yolk, approximately 50-100 μL collected with insulin syringes. The embryo was transferred to a sterile dissecting plate and dissected to sample the GI tract. Each embryo was dissected with new and sterile dissection plates, gloves, and sterile surgical instruments. All embryonic samples were placed and stored in sterile propylene tubes at -20°C until their subsequent DNA extraction within five days. Only samples that were free from cross-contamination, such as membrane with yolk, were considered for analysis. As a result, we were able to retrieve six membrane samples, four yolk samples, and six amniotic fluid samples out of a total of 17 embryos.”

“Blank controls were included in each extraction batch. Blank controls were verified for contamination by electrophoresis and amplification of the 16S rRNA gene.”

“Blank, positive and reagent controls were included in each PCR run. All these controls were included in the amplification process with the sequencing adapters, but no amplicons were obtained. Sequencing was done by MacroGen Inc using the Illumina MiSeq platform and 300-bp PE runs (Illumina, San Diego, CA, USA). DNA Sequencing Service (Seoul, South Korea). The raw sequences were analyzed to identify human DNA sequences, which could serve as potential indicators of contamination during the sequencing process. No human DNA was detected.”

Functional Prediction:

• *The use of PICRUSt analysis to predict the potential functions of the bacterial microbiota is appropriate.*

Thank you for your comment.

• *Elaborate on the functional implications of the observed differences in potential functions between embryonic and maternal tissues.*

We included more information on the potential functions of the embryonic and maternal tissues as suggested by the reviewer.

“The most abundant functions in the embryonic samples were related to aromatic compounds and amino acids degradation, e.g. leucine and 4-aminobutanoate degradation, pyruvate fermentation, and NAD and heme biosynthesis. The most abundant functions in maternal tissues were related to cofactor, carrier and vitamin biosynthesis, fermentation, and carbohydrate degradation, such as fucose and rhamnose degradation, menaquinol and demethylmenaquinol biosynthesis, dTDP-N-acetylthomosamine biosynthesis, besides pyruvate and succinate fermentation (Fig. 5). The CMP-legionaminatate (5,7-diacetamido-3,5,7,9-tetrahydroxy-D-glycero-D-galactononulosonic acid) biosynthesis I was exclusive in the embryonic and maternal GI tract, while other functions exclusively found in the embryonic GI tract were involved in the degradation of aromatic compounds, such as mandelate, arabinose and catechol.”

“We investigated the potential functions of bacterial communities in both the embryo and the mother to understand their possible roles. Predicted functions related to the degradation of mucin, such as fucose and rhamnose degradation pathways, were more abundant in the maternal samples (mouth, cloaca and small intestine). This suggests the presence of bacteria that utilize mucin as a carbon source (84). Additionally, functions related to carbohydrate metabolism indicate the presence of strictly anaerobic bacteria in the maternal intestine and aerobic or facultative anaerobic bacteria in the embryonic GI tract (85). Another putative function that was more abundant in the maternal samples was related to the biosynthesis of vitamin K, specifically “the biosynthesis superpathway of menaquinol and demethylmenaquinol”. Vitamin K is a secondary metabolite produced by mutualistic bacteria, highlighting the association of the bacterial microbiota with its host and showing the more complex bacterial community in the mother compared to that in the embryo. Additionally, pyruvate and succinate fermentation pathways were more abundant in maternal samples than in embryonic samples. These pathways produce short-chain fatty acids, such as acetate, propionate and butyrate (86). These metabolites can be used by the maternal intestinal epithelial cells as an energy source and may contribute to cell differentiation of the embryonic epithelium (87, 88).

Pathways related to the biosynthesis of cofactors and vitamins, such as NAD and heme group biosynthesis, were abundant in embryonic samples. These molecules are acceptors or donors of electrons and recover energy metabolism, and participate in glycolysis, gluconeogenesis and oxidative stress (89). These pathways might play an important role in obtaining energy for the developing embryo and generate a redox balance in the amniotic environment.

A function related with metabolism L-leucine was very abundant in the embryonic and maternal bacterial communities. Leucine has been identified as an important precursor for the biosynthesis of fatty acids in embryonic tissues (90). Fatty acids and proteins are significant constituents of embryonic tissues mainly in the yolk (91). During development, the mother supplies precursors for these components to the embryo

through the fetal membranes, which could explain the high abundance of this function in both the mother and the embryo. The CMP-legionamate biosynthesis was exclusive in the embryonic and maternal intestine. This function is implicated in the immune system modulation (92). The bacterial processes involved in the health and development of the host occur in the small intestine, e.g. the maturation of the immune system. This might indicate that the bacteria-intestine interaction is participating in the maturation of the immune system of the developing embryos of the lizard *S. grammicus*.”

• *Discuss any functional patterns or pathways that are unique to specific sample sites or stages of development.*

We added this information to the discussion section as the reviewer recommends.

“The CMP-legionamate biosynthesis was exclusive in the embryonic and maternal intestine. This function is implicated in the immune system modulation (92). The bacterial processes involved in the health and development of the host occur in the small intestine, e.g. the maturation of the immune system. This might indicate that the bacteria-intestine interaction is participating in the maturation of the immune system of the developing embryos of the lizard *S. grammicus*.”

Overlap between Embryonic and Maternal Microbiotas:

• *The comparison of ASVs between embryonic and maternal samples provides valuable insights into the potential transfer of microbiota.*

Thank you for your comment.

• *Discuss the biological significance of the overlapping ASVs and the implications for vertical transmission of the microbiota.*

We added more information related to the biological significance of the overlapping ASVs and the implications for vertical transmission of the microbiota as the reviewer suggests.

“It is known that human babies delivered by C-section harbor bacteria in their intestines similar to those found on the maternal skin (8). While the exact mechanism of bacterial transfer from maternal skin to the newborn's intestines is still debated, the prevailing hypothesis suggests that skin bacteria are transmitted during skin-to-skin interactions between the newborn and the mother or medical personnel (60). In our study, we observed a high similarity between the bacteria present on the maternal skin and those found in the embryos, despite the embryos never coming into direct contact with skin tissues. Our findings indicate that skin bacteria may potentially be transmitted before birth and influence vertebrate development.

The high similarity of maternal oral bacteria with bacteria in the embryo has also been detected in human maternal transmission studies (40, 61). The role and biological implication of these oral commensal bacteria in maternal transmission is still unknown. One proposed explanation is that oral bacteria facilitate hematogenous transmission by binding to the vascular endothelium and altering its permeability, thus functioning as "enabler" so that other commensal bacteria can reach the placental environment (40). This ability to destroy the integrity of the endothelium was attributed to *Fusobacterium nucleatum* (62). This bacterium commonly identified in the mouth contains adhesins that bind to vascular endothelial cadherin.”

Overall, the "Results" section provides detailed information on the taxonomic and functional composition of embryonic and maternal microbiotas. However, it would be beneficial to expand on the interpretation and biological relevance of the findings, consider discussion of

vertical transmission contamination on human's motherhood to the fetus is often by physical rupture or through blood circulation.

We added more information to the text and discussed this as suggested by the reviewer.

“The high similarity of maternal oral bacteria with bacteria in the embryo has also been detected in human maternal transmission studies (40, 61). The role and biological implication of these oral commensal bacteria in maternal transmission is still unknown. One proposed explanation is that oral bacteria facilitate hematogenous transmission by binding to the vascular endothelium and altering its permeability, thus functioning as "enabler" so that other commensal bacteria can reach the placental environment (40). This ability to destroy the integrity of the endothelium was attributed to *Fusobacterium nucleatum* (62). This bacterium commonly identified in the mouth contains adhesins that bind to vascular endothelial cadherin.”

Reviewer #2 (Comments for the Author):

Regarding the microbial transmission from mother to child, due to sampling difficulties and experimental contamination problems, no more definite conclusions have been drawn so far. In this study, this study innovatively used viviparous lizard as the study subject and observed the maternal microbial transmission during embryonic development.

There are two minor issues:

1. In the analysis of maternal microbial transmission, the "betapart" method was used, while in other microbial traceability studies, methods such as "Sourcetracker" or "FEAST" are often used. What is the basis for the authors' choice of the former? What is the difference between the former and the latter? Which method is superior?

We chose the betapart method because it is a valuable tool in community ecology studies (Baselga and Orme 2012). The analysis is based on the partitioning of beta diversity into two components: turnover and nestedness. In our study, the nestedness component was useful to test whether the embryonic microbiota was a subset of the maternal microbiota and to trace the possible origin of the bacteria detected in the embryo. We agree with the reviewer that there are other powerful and widely used tools that give similar results. FEAST program partitions microbial samples into their source components. FEAST uses an Expectation-Maximization algorithm to determine the contribution ratios of known and unknown sources to data sinks. Through successive iterations, gradually adjusts the estimates to achieve a solution that best fits the observed data (Shenhav et al. 2019). We used FEAST, but the program did not allow us to include different sources (different maternal samples) and sinks (different embryonic samples) at the same time. In addition, the program coerces the contribution from different sources to fixed proportions, which does not allow analyzing a possible overlapping of microbiotas. We presumed that the microorganisms present in the different anatomical parts are not necessarily unique in a certain region. We were also interested in knowing the coincidences between different samples. However, to make our results comparable with studies using other tools of analysis (i.e. FEAST) and support the possible maternal source of bacteria in the embryos, we included the FEAST analysis in the manuscript.

“We determined the possible maternal origin of bacteria in the embryos with FEAST (fast expectation-maximization for microbial source tracking) analysis. Overall, we found high source contributions of maternal mouth (mean 28.2%) and aseptic ventral skin (mean 27.8%), and from unknown sources (mean 45.4%; Fig. S2). Maternal cloaca and small intestine were not a significant potential source of bacteria. Over half of the

bacterial composition in the embryonic GI tract (54.8%) originated from unknown sources. This result is consistent with the significant presence of unique ASVs in the embryonic samples.”

“FEAST analysis (123) was done to determine the contribution of the different maternal sources to the bacteria in the embryos. The analysis was done with each of the mothers and their respective embryos. Maternal samples were defined as sources and embryonic samples as sinks.”

Baselga A, Orme CDL. 2012. betapart: an R package for the study of beta diversity. *Methods Ecol Evol* 3:808-812.

Shenhav L, Thompson M, Joseph TA, Briscoe L, Furman O, Bogumil D, Mizrahi I, Pe'er I, Halperin E. 2019. FEAST: rapid maximization of expectations for the monitoring of microbial sources. *Nat Methods* 16:627-632.

2, the horizontal coordinates of Figure 7, the different color bars, I guess should be "the amniotic fluid, embryonic gastrointestinal (GI) tract, membrane and yolk of 40 stage embryos", these should be specifically marked out

The caption was included at the top of the figure, but we have now added this annotation to the horizontal axis to avoid confusion.

Reviewer #3 (Comments for the Author):

In this study, Montoya-Ciriaco et al, study the transmission of maternal microbiota to embryos. They use high throughput sequencing of the 16s rRNA bacterial genes to evaluate the relative composition of embryo and amniotic fluid bacteria compared to maternal tissues. The authors find reduced diversity in the bacterial composition in embryos compared to maternal tissue suggesting a controlled vertical microbiota transmission. The authors propose that maternal microbiota was transmitted, early in embryo development, mainly from the mouth and aseptic ventral skin to the embryos based on amplicon sequence variants. The field is still debating whether maternal-fetal microbiota transfer occurs early in embryo although emerging lines of evidence appear to support this hypothesis. A lingering concern in the field has been the possibility of low-level contamination in the samples. The authors try to address this concern in the paper. Overall, the study is straightforward and the reduced diversity in the bacterial population in embryos matches reports from human newborn meconium. The data supports the hypothesis, although more work would be needed to definitely point to the importance and role of the microbiota in reptile development, which the authors refer to but do not address in this study. In addition, where and how the selection/filtration of bacterial species occur still remains elusive. I have a few comments that are detailed below:

We appreciate all the comments made by the reviewer which helped us to improve our manuscript.

- It seems that the bacteria that are relatively low abundance in the mother are being transmitted disproportionately to the embryo. It also appears to be higher than other maternal tissues in the control. The authors mention the relative abundance in figure 2 but how different is the total amount of bacteria in the different samples? Can this be estimated from amplification amplitude? Can the authors comment on this?*

It is an interesting question. The data generated by sequencing are not the real or true counts of the bacterial communities, so the relative abundance does not represent the total number of bacteria. It is not possible to determine the number of bacteria by the amplification amplitude as long as a quantitative technique is not applied. Sometimes it is possible to do it by metabarcoding when the primers amplify the host mitochondrial DNA well because you can see per sample the ratio of host vs bacteria sequences. However, this was not the case in this study.

• *It is also curious that the pattern of the 50 most abundant ASVs is most similar to the mouth and the non-superficial bacteria in the control (aseptic ventral skin). That is puzzling. Are the bacteria closer to the epithelium just harder to eliminate and perhaps overrepresented in samples somehow?*

We agree with the reviewer and presume that the bacteria closely attached to the skin epithelium are more difficult to eliminate. When performing the aseptic technique on the ventral body region, the more superficial bacteria on the epidermis and the lizard's stratum corneum were eliminated, the diversity was reduced and those ASVs represented in the heatmap are the bacteria related to the dermis, the inner and deeper layer of the skin. We did not refer to them as skin bacteria, because it was modified by the aseptic technique. Still some bacteria were detected, and similar bacteria were also detected in the embryonic samples. We tried to clarify this in the manuscript.

“It is known that human babies delivered by C-section harbor bacteria in their intestines similar to those found on the maternal skin (8). While the exact mechanism of bacterial transfer from maternal skin to the newborn's intestines is still debated, the prevailing hypothesis suggests that skin bacteria are transmitted during skin-to-skin interactions between the newborn and the mother or medical personnel (60). In our study, we observed a high similarity between the bacteria present on the maternal skin and those found in the embryos, despite the embryos never coming into direct contact with skin tissues. Our findings indicate that skin bacteria may potentially be transmitted before birth and influence vertebrate development.”

• *The authors mention 3 ways of transferring bacteria to the embryo - placenta, vaginal spread, intestinal absorption. Transplacental trafficking has been suggested widely. The amniotic fluid looks very similar to other embryo tissues, so the maternal control in terms of which bacteria are transmitted, would have occurred upstream. The membrane and the yolk share a lot of ASVs with the mother but the selection has already occurred given the ASVs still belong to a few taxonomic groups. The filtration/selection could happen when passing through the membrane but I find it hard to envision a species-specific filtration system for microbiota in the membrane.*

How would that work?

We hypothesize that it may be via antimicrobial peptides (AMPs) because these metabolites prevent colonization of Gram-positive bacteria and most of the bacteria detected in the embryonic samples were Gram-negative. This mechanism has been widely studied in the model organism, the Hawaiian bobtail squid. The selection occurs while the hatchling organ of the squid is developing. A similar mechanism with the membrane and the yolk can occur and the selection process started with the development of these structures. However, the reviewer is right and the hypothesis can be complemented with the transplacental trafficking observed in humans. We included this in the discussion.

“The most abundant ASVs in the embryos of *S. grammicus* were mostly Gram-negative bacteria (see Fig. 3). So, one hypothesis might be that during the formation of the extraembryonic membranes and yolk these structures produced AMPs that filter or select specific bacteria.

In placental organisms, such as humans, transplacental trafficking has been proposed as a route of cell exchange between mother-fetus. This mechanism could be involved in the selection of bacteria in the maternal placenta and in the fetal membranes of the embryo. The extraembryonic membrane is closely linked to the *S. grammicus* placenta and contains trophoblasts and specialized cells similar to those present in the human placenta (29, 99). These epithelial cells participate in bacterial selection and/or pathogen defense via AMPs, defensins and Toll-like receptors (50, 52). As such, a second hypothesis, which does not exclude the previous one, is that given the placental function as the interface between the maternal and fetal environment, it is possible that the placenta constitutes the first filter between the maternal and fetal environment. Nonetheless, the exact mechanisms governing the restriction and/or promotion of microbial access remain undefined (50).”

Villagrán Santa Cruz M. 1989. “Desarrollo embrionario placentación y su relación con el cuerpo lúteo y la atresia folicular en *Sceloporus mucronatus* y *Sceloporus grammicus*, (Sauria: Iguanidae)”. (Tesis de Doctorado). Universidad Nacional Autónoma de México, México.

Thompson MB, Biazik JB, Lui S, Adams SM, Murphy CR. 2010. Morphological and functional changes to the uterus of lizards with different placental complexities. *Herpetol Monogr* 20:178-185.

King AE, Paltoo A, Kelly RW, Sallenave JM, Bocking AD, Challis JRG. 2007. Expression of natural antimicrobials by human placenta and fetal membranes. *Placenta* 28:161-169.

Tangerås LH, Stødle GS, Olsen GD, Leknes AH, Gundersen AS, Skei B, Vikdal AJ, Ryan L, Steinkjer B, Myklebost MF, Langaas M, Austgulen R, Iversen AC. 2014. Functional Toll-like receptors in primary first-trimester trophoblasts. *J Reprod Immunol* 106:89-99.

Arora N, Sadovsky Y, Dermody TS, Coyne CB. 2017. Microbial vertical transmission during human pregnancy. *Cell Host Microbe* 21:561-567.

• Is the diversity in pregnant female lizards different than non-pregnant or nulliparous ones? This would be interesting in light of the fact that proteobacteria tend to increase upon pregnancy. It is possible that there is a pre-defined bias very early in pregnancy that sets the system up for a biased transmission of certain bacteria. This is highlighted by the 28% of ASVs in the embryonic GI tract that are not shared with any maternal sample. The authors mention that these may come from earlier gravid stages and again that comes back to investigating time resolved maternal microbiota distribution during gestation. It is possible that the maternal microbiota is already being selectively shaped for vertical transmission.

We agree with the reviewer and are to investigate this in other studies.

• A minor comment would be to tone down the language for whether the bacterial transmission is crucial for lizard development and adaptation in the abstract as none of that was tested in this study.

The reviewer is right, and we changed the text in the abstract section.

“Our study provides evidence of microbiota vertical transfer during embryonic development in the animal kingdom. It also highlights that this maternal transmission could be included in the maternal effects that impact the offspring.”

September 8, 2023

Dr. Yendi E Navarro-Noya
Universidad Autónoma de Tlaxcala
Centro de Investigación en Ciencias Biológicas
Tlaxcala, Tlaxcala 90000
Mexico

Re: Spectrum01780-23R1 (**Maternal transmission of bacterial microbiota during embryonic development in a viviparous lizard**)

Dear Dr. Yendi E Navarro-Noya:

Your manuscript has been accepted, and I am forwarding it to the ASM Journals Department for publication. You will be notified when your proofs are ready to be viewed.

Sincerely,

Diyan Li
Editor, Microbiology Spectrum

Journals Department
From,

Dr Avvari Bhaskara Balaji

Sr. Scientist

Prof. G. M. Reddy Research Foundation

& SSSA, India

Date: 3-09-2023

To,

Dr Diyan Li

Dear Diyan Li,

I have had the opportunity to review the revised manuscript titled " Maternal transmission of bacterial microbiota during embryonic development in a viviparous lizard " submitted by Yendi Navarro-Noya for possible publication in Microbiology Spectrum. I appreciate the authors' thorough response to the previous comments and suggestions, and I have reevaluated the manuscript based on the revisions made. Below, I provide a summary of changes of the revised manuscript and my justification for the authors' work.

Summary of changes of Revised Manuscript and its justification:

Reviewer 1

Clarity and Organization:

Sample Description and Sequencing Methods: The description of the samples analyzed and the sequencing methods used has been clarified and organized for better comprehension of the statistical analysis and the study's scope. Specific information about the number of samples analyzed for each category has been incorporated.

Alpha Diversity:

The reviewer's affirmation that Hill numbers are appropriate for analyzing alpha diversity has been acknowledged, and their comments regarding the suitability of this approach have been incorporated into the manuscript.

Differences in Bacterial Diversity: In response to valuable feedback, the discussion on the differences in bacterial diversity between maternal and embryonic samples has been expanded. This comprehensive explanation now elucidates the potential reasons behind these differences, adding substantial value to the manuscript by highlighting the significance of the findings.

Taxonomic Composition:

A detailed description of the differences in taxonomic composition between different sample sites within the maternal samples has been provided. This contextualizes and provides insights into the microbial communities in various regions of the maternal lizard.

Community Composition and Structure:

Specific amplicon sequence variants (ASVs) in both maternal and embryonic samples have been identified, and a brief discussion of the potential roles or functions of these identified ASVs has been included. This enriches the analysis and helps readers understand the relevance of these microbial taxa in the studied environments.

In addition to these revisions, enhancements have been made in response to the reviewer's comments on the "Results" and "Materials and Methods" sections:

PCA and perMANOVA Interpretation: The interpretation of the PCA and perMANOVA results has been expanded, providing a clear description of clustering patterns and their implications for better reader comprehension of the observed differences in bacterial communities between embryonic and maternal samples.

Contamination Controls: A detailed description of the steps taken to minimize or control for contamination during sample collection and sequencing has been provided in the "Materials and Methods" section. This transparency underscores the rigor of the experimental procedures and effectively addresses the reviewer's concern.

Functional Predictions: The manuscript now includes elaboration on the functional implications of observed differences in potential functions between embryonic and maternal tissues. This deepens the understanding of the findings and sheds light on the potential roles of bacterial communities in different sample types.

Unique Functional Patterns: Functional patterns or pathways that are unique to specific sample sites or stages of development have been discussed, providing further insights into the functional differences between embryonic and maternal microbiotas.

Overlap between Embryonic and Maternal Microbiotas: The biological significance of overlapping ASVs and their potential implications for vertical transmission of the microbiota has been explored. This connection to broader biological concepts and hypotheses strengthens the impact of the findings.

Overall, I find that these revisions have significantly improved the quality and comprehensibility of the manuscript, and I recommend its acceptance for publication. The authors have addressed all previous concerns effectively, and the manuscript now represents a valuable contribution to the field.

Thank you for considering my evaluation of the revised manuscript.

Sincerely,

Dr Avvari Bhaskara Balaji

Mobile No: 91+8463978298